# Fast Estimation of Wasserstein Distances via Regression on Sliced Wasserstein Distances

**Khai Nguyen**[*]
Department of Statistics and Data Sciences
University of Texas at Austin
Austin, TX 78713, USA
`khainb@utexas.edu`

**Hai Nguyen**[*]
Independent Researcher
`namhai283287@gmail.com`

**Nhat Ho**
Department of Statistics and Data Sciences
University of Texas at Austin
Austin, TX 78713, USA
`minhnhat@utexas.edu`

## Abstract

We address the problem of efficiently computing Wasserstein distances for multiple pairs of distributions drawn from a meta-distribution. To this end, we propose a fast estimation method based on regressing Wasserstein distance on sliced Wasserstein (SW) distances. Specifically, we leverage both standard SW distances, which provide lower bounds, and lifted SW distances, which provide upper bounds, as predictors of the true Wasserstein distance. To ensure parsimony, we introduce two linear models: an unconstrained model with a closed-form least-squares solution, and a constrained model that uses only half as many parameters. We show that accurate models can be learned from a small number of distribution pairs. Once estimated, the model can predict the Wasserstein distance for any pair of distributions via a linear combination of SW distances, making it highly efficient. Empirically, we validate our approach on diverse tasks, including Gaussian mixtures, point-cloud classification, and Wasserstein-space visualizations for 3D point clouds. Across various datasets such as MNIST point clouds, ShapeNetV2, MERFISH Cell Niches, and scRNA-seq, our method consistently provides a better approximation of Wasserstein than the state-of-the-art method, Wasserstein Wormhole, and classical methods, particularly in low-data regimes. To illustrate its robustness, we also experiment the method with intra- and inter-class settings. Finally, we demonstrate that *RG* can accelerate Wasserstein Wormhole training, yielding *RG-Wormhole*. [1]

## 1    Introduction

Optimal Transport (OT) and Wasserstein distances (Villani, 2009; Peyré & Cuturi, 2019) have become essential tools in machine learning, widely used for quantifying the similarity or dissimilarity between probability distributions. Fundamentally, the Wasserstein distance measures the minimum cost required to "transport" mass from one distribution to another, effectively capturing the underlying geometry of the data. Thanks to their clear geometric interpretation and mathematical robustness, Wasserstein distances have found applications across various fields, such as generative modeling Genevay et al. (2018), computational biology Bunne et al. (2023), chemistry Wu et al. (2023), and image processing Feydy et al. (2017). Despite its utility, computing the exact Wasserstein distance is computationally expensive. It typically requires solving a large-scale linear program to find an optimal transport plan, with a time complexity of $\mathcal{O}(n^3 \log n)$ for discrete distributions of size $n$. This high cost severely limits its use in large-scale or real-time settings.

---

[*]Equal Contribution
[1]Code is published at `https://github.com/hainn2803/Regression-Wasserstein`.

In many applications, Wasserstein distances are computed (repeatedly) for many pairs of distributions instead of a single pair, e.g., dataset comparisons (Alvarez-Melis & Fusi, 2020), 3D point-cloud autoencoder (Achlioptas et al., 2018), point-cloud nearest neighbor classification/regression (Rubner et al., 1998), learning embeddings for distributions (Kolouri et al., 2021), density-density regression (Chen et al., 2023), and so on. Therefore, the high computational complexities of the Wasserstein distance become the main bottleneck to scaling up these applications. As a result, speeding up the computation of the Wasserstein distance has become a vital task in practice.

To address this bottleneck, a straightforward improvement is to speed up the computation of the Wasserstein distance. For example, entropic regularization (Cuturi, 2013) enables fast approximation via Sinkhorn iterations, while other methods exploit the structure in the transport plan, such as low-rank approximations (Scetbon et al., 2021). In addition, some approaches rely on strong structural assumptions, such as the Bures-Wasserstein metric (Dowson & Landau, 1982) gives a closed-form solution for the exact 2-Wasserstein distance ($W_2$) under the Gaussian assumption on distributions.

Another approach is to cast computing Wasserstein distances for many pairs of distributions as a learning problem, i.e., learning a model first to predict the Wasserstein distance given any pair of distributions, then use the model later for the mentioned downstream tasks. For example, Deep Wasserstein Embedding (DWE) (Courty et al., 2018) trains a Siamese convolutional network to match OT distances between 2D images, while Wasserstein Wormhole (Haviv et al., 2024) employs transformer-based architectures to learn embeddings of distributions, allowing Euclidean distances in the learned space to approximate Wasserstein distances efficiently. While effective, these deep learning-based methods require significant computational resources and time to train, and their performance may degrade when limited training data are available. Moreover, these approaches are limited to empirical distributions because of the use of neural networks.

In this work, we propose a novel approach to predict the Wasserstein distance without relying on any neural networks or learned embeddings. Moreover, the proposed approach relies on a parsimonious model and can handle both continuous and discrete distributions. In particular, we propose to regress the Wasserstein distance on sliced Wasserstein (SW) distances (Rabin et al., 2010; Mahey et al., 2023; Nguyen & Ho, 2023; Liu et al., 2025; Deshpande et al., 2019; Rowland et al., 2019). In greater detail, we introduce linear models with Wasserstein distances as the response and SW distances as the predictors. We provide estimates of the models via efficient least-squares estimates. In addition, since sliced Wasserstein distances have low computational complexity, the resulting Wasserstein regressor is computationally efficient.

**Contribution:** In summary, our main contributions are three-fold:

1. We introduce the first regression framework where the Wasserstein distance serves as the response variable and various sliced Wasserstein (SW) distances act as predictors, in the setting of random pairs of distributions. This framework not only uncovers the relationship between the Wasserstein distance and its SW-based approximations but also enables efficient estimation of the Wasserstein distance. Specifically, we use SW distance (Bonneel et al., 2015), Max-SW (Deshpande et al., 2019), and energy-based SW (Nguyen & Ho, 2023), all of which provide lower bounds on the Wasserstein distance, as predictors. In addition, we incorporate lifted SW distances, which provide upper bounds, including projected Wasserstein (Rowland et al., 2019), minimum SW generalized geodesics (Mahey et al., 2023), and expected sliced distance (Liu et al., 2025).

2. We propose two linear models for the regression problem and describe their estimation via least-squares. The first model is unconstrained and admits a closed-form least-squares solution. The second model incorporates constraints that leverage the known bounds between SW distances and the Wasserstein distance, thereby reducing the number of parameters by half. Based on these estimations, we obtain a fast method to approximate the Wasserstein distance for any pair of distributions, with the same computational complexity as that of computing SW distances.

3. Empirically, we demonstrate that our approach yields accurate estimates of the Wasserstein distance, particularly in low-data regimes. We first evaluate its accuracy through simulations with Gaussian mixtures. We then apply the estimated distances to visualize distributional data and to perform $k$-NN classification on ShapeNetV2 point clouds. Next, we benchmark our method against Wasserstein Wormhole, the state-of-the-art Wasserstein embedding model, across four datasets of increasing dimensionality: MNIST point clouds, ShapeNetV2, MERFISH cell niches, and scRNA-seq.

Finally, we propose *RG-Wormhole*, a variant of Wasserstein Wormhole that replaces its Wasserstein computations with our estimates, preserving accuracy while substantially reducing training time.

**Organization.** Section 2 reviews preliminaries on the Wasserstein distance, its sliced variants, and their computation. Section 3 introduces our regression framework for approximating Wasserstein distances from sliced variants, together with both constrained and unconstrained linear models. Section 4 reports the experimental results. The appendices provide supplementary experiments (mixtures of Gaussians and distributional space visualizations), detailed experimental settings, theoretical proofs, and additional related work.

**Notations.** For any $d \geq 2$, let $\mathbb{S}^{d-1} := \{\theta \in \mathbb{R}^d : \|\theta\|_2 = 1\}$ denote the unit sphere in $\mathbb{R}^d$, and let $\mathcal{U}(\mathbb{S}^{d-1})$ denote the uniform distribution on it. For $p \geq 1$, we write $\mathcal{P}_p(\mathcal{X})$ for the set of all probability measures on $\mathcal{X}$ with the finite $p$ th moment. Given two sequences $a_n$ and $b_n$, the notation $a_n = \mathcal{O}(b_n)$ means that $a_n \leq Cb_n$ for all $n \geq 1$, for some universal constant $C > 0$. For a measurable map $P$, the notation $P\sharp\mu$ denotes the push-forward of $\mu$ through $P$. Additional notation will be introduced as needed.

## 2 PRELIMINARIES

We first review definitions and computational aspects of the Wasserstein distance and its related properties in one dimension.

**Wasserstein distance.** Wasserstein-$p$ ($p \geq 1$) distance Villani (2008); Peyré et al. (2019) between two distributions $\mu \in \mathcal{P}_p(\mathbb{R}^d)$ and $\nu \in \mathcal{P}_p(\mathbb{R}^d)$ (dimension $d \geq 1$) is defined as:

$$W_p^p(\mu, \nu) = \inf_{\pi \in \Pi(\mu,\nu)} \int_{\mathbb{R}^d \times \mathbb{R}^d} \|x - y\|_p^p d\pi(x, y), \tag{1}$$

where $\Pi(\mu, \nu) = \left\{\pi \in \mathcal{P}(\mathbb{R}^d \times \mathbb{R}^d)\right\} \mid \int_{\mathbb{R}^d} d\pi(x, y) = \mu(x), \int_{\mathbb{R}^d} d\pi(x, y) = \nu(y)\right\}$ is the set of all transportation plans i.e., joint distributions which have marginals be two comparing distributions. When $\mu$ and $\nu$ are discrete distributions i.e., $\mu = \sum_{i=1}^n \alpha_i \delta_{x_i}$ ($n \geq 1$) and $\nu = \sum_{j=1}^m \beta_j \delta_{y_j}$ ($m \geq 1$) where $\sum_{i=1}^N \alpha_i = \sum_{j=1}^m \beta_j = 1$ and $\alpha_i \geq 0, \beta_j \geq 0$ for all $i = 1, \ldots, n$ and $j = 1, \ldots, m$, Wasserstein distance between $\mu$ and $\nu$ defined as:

$$W_p^p(\mu, \nu) = \min_{\gamma \in \Gamma(\alpha,\beta)} \sum_{i=1}^n \sum_{j=1}^m \|x_i - y_j\|_p^p \gamma_{ij},$$

where $\Gamma(\alpha, \beta) = \{\gamma \in \mathbb{R}_+^{n \times m} \mid \gamma\mathbf{1} = \alpha, \gamma^\top\mathbf{1} = \beta\}$. Without loss of generality, we assume that $n \geq m$. Therefore, the time complexity for solving this linear programming is $\mathcal{O}(n^3 \log n)$ Peyré & Cuturi (2019) and $\mathcal{O}(n^2)$, which are expensive.

**One-dimensional Case.** When $d = 1$, the Wasserstein distance can be efficiently calculated. For the continuous case, Wasserstein-2 distance has the following form: $W_p^p(\mu, \nu) = \int_0^1 |F_\mu^{-1}(t) - F_\nu^{-1}(t)|^p dt$, where $F_\mu^{-1}$ and $F_\nu^{-1}$ denote the quantile functions of $\mu$ and $\nu$ respectively. Here, the transportation plan is $\pi_{(\mu,\nu)} = (F_\mu^{-1}, F_\nu^{-1})\sharp\mathcal{U}([0, 1])$. When $\mu$ and $\nu$ are discrete distributions, i.e. $\mu = \sum_{i=1}^n \alpha_i \delta_{x_i}$ ($n \geq 1$) and $\nu = \sum_{j=1}^m \beta_j \delta_{y_j}$, quantile functions of $\mu$ and $\nu$ are:

$$F_\mu^{-1}(t) = \sum_{i=1}^n x_{(i)} I\left(\sum_{j=1}^{i-1} \alpha_{(j)} < t \leq \sum_{j=1}^i \alpha_{(j)}\right), F_\nu^{-1}(t) = \sum_{j=1}^m y_{(j)} I\left(\sum_{i=1}^{j-1} \beta_{(i)} < t \leq \sum_{i=1}^j \beta_{(i)}\right),$$

where $x_{(1)} \leq \ldots \leq x_{(n)}$ and $y_{(1)} \leq \ldots \leq y_{(m)}$ are the sorted supports (or order statistics). Therefore, the one-dimensional Wasserstein distance can be computed in $\mathcal{O}(n \log n)$ in time and $\mathcal{O}(n)$ in space (assuming that $n > m$).

**Random Projection.** A key technique that plays a vital role in later discussion is random projection. We consider a function $P_\theta : \mathbb{R}^d \to \mathbb{R}$ where $\theta \sim \sigma(\theta)$ ($\sigma(\theta) \sim \mathcal{P}(\mathbb{S}^{d-1})$) is a random variable. For simplicity, we consider the traditional setup where $\theta \sim \mathcal{U}(\mathbb{S}^{d-1})$ and $P_\theta(x) = \langle\theta, x\rangle$ (Bonneel et al., 2015; Rabin et al., 2012). However, the following discussion holds for any other types of

projections (Kolouri et al., 2019; Bonet et al., 2023b; 2025; 2023c). For $\mu \in \mathcal{P}_p(\mathbb{R}^d)$ and $\nu \in \mathcal{P}_p(\mathbb{R}^d)$, one-dimensional projected Wasserstein distance with $P_\theta$ is defined as:

$$\underline{W}_p^p(\mu, \nu; P_\theta) = W_p^p(P_\theta \sharp \mu, P_\theta \sharp \nu) = \int_0^1 |F_{P_\theta \sharp \mu}^{-1}(t) - F_{P_\theta \sharp \nu}^{-1}(t)|^p dt. \tag{2}$$

The second approach to construct a Wasserstein-type discrepancy from one-dimensional projection is using lifted transportation plan. There are many ways to construct such lifted plan using disintegration of measures (Muzellec & Cuturi, 2019; Tanguy et al., 2025). In practice, the most used way (Liu et al., 2025; Tanguy et al., 2025) is:

$$\overline{W}_p^p(\mu, \nu; P_\theta) = \int_{\mathbb{R}^d \times \mathbb{R}^d} \|x - y\|_p^p d\pi^\theta(x, y) \tag{3}$$

$$= \int_{\mathbb{R} \times \mathbb{R}} \int_{P_\theta^{-1}(t_1) \times P_\theta^{-1}(t_2)} \|x - y\|_p^p d\mu_{t_1} \otimes \nu_{t_2}(x, y) d\pi_{(P_\theta \sharp \mu, P_\theta \sharp \nu)}(t_1, t_2), \tag{4}$$

where $\pi^\theta \in \Pi(\mu, \nu)$ is the lifted transportation plan, $\pi_{(P_\theta \sharp \mu, P_\theta \sharp \nu)}$ is the optimal transport plan between $P_\theta \sharp \mu$ and $P_\theta \sharp \nu$, $\mu_{t_1}$ and $\nu_{t_2}$ are disintegration of $\mu$ and $\nu$ at $t_1$ and $t_2$ the function $P_\theta$, and $\otimes$ denotes the product of measures. When dealing with discrete measures $\mu$ and $\nu$, $\overline{W}_p^p(\mu, \nu; P_\theta)$ can still be computed efficiently (Mahey et al., 2023; Liu et al., 2025) i.e., $\mathcal{O}(n \log n)$ in time and $\mathcal{O}(n)$ in space (assumed that $n > m$). The quantity $\overline{W}_p^p(\mu, \nu; P_\theta)$ is known as lifted cost (Tanguy et al., 2025) or sliced Wasserstein generalized geodesic (Mahey et al., 2023; Liu et al., 2025). From previous work (Nguyen & Ho, 2023; Mahey et al., 2023; Tanguy, 2023), we know the following relationship $\underline{W}_p(\mu, \nu; P_\theta) \leq W_p(\mu, \nu) \leq \overline{W}_p(\mu, \nu; P_\theta)$.

# 3 REGRESSION OF WASSERSTEIN DISTANCE ONTO SLICED OPTIMAL TRANSPORT DISTANCES

In this section, we present a framework for regressing the Wasserstein distance onto sliced Wasserstein distances, propose some models, and discuss related computational properties.

## 3.1 SLICED WASSERSTEIN AND LIFTED SLICED WASSERSTEIN

**Sliced Wasserstein distances.** Given $\mu \in \mathcal{P}_p(\mathbb{R}^d)$ and $\nu \in \mathcal{P}_p(\mathbb{R}^d)$, a sliced Wasserstein-$p$ distance can be defined as follows (Rabin et al., 2012; Nguyen, 2025):

$$SW_p^p(\mu, \nu; \sigma) = \mathbb{E}_{\theta \sim \sigma} \left[ \underline{W}_p^p(\mu, \nu; P_\theta) \right], \tag{5}$$

where $P_\theta : \mathbb{R}^d \to \mathbb{R}$ is the projection function, $\underline{W}_p^p(\mu, \nu; P_\theta)$ is the one-dimensional projected Wasserstein distance (equation 2), and $\sigma \in \mathcal{P}(\mathbb{S}^{d-1})$ is the slicing distribution. By changing the slicing distribution, we can obtain variants of SW. There are three main ways:

1. *Fixed prior:* The simplest way is to choose $\sigma$ to be a fixed and known distribution, e.g., the uniform distribution $\mathcal{U}(\mathbb{S}^{d-1})$ as in the conventional SW (Rabin et al., 2012).

2. *Optimization-based:* We can also find $\sigma$ that prioritizes some realizations of $\theta$ that satisfies a notion of informativeness. For example, $\sigma$ can put more masses to realizations of $\theta$ where $\underline{W}_p^p(\mu, \nu; P_\theta)$ have high value, i.e., setting informativeness as discriminativeness. For example, we can find $\sigma$ by solving (Nguyen et al., 2021):

$$\sup_{\sigma \in \mathcal{M}(\mathbb{S}^{d-1})} \mathbb{E}_{\theta \sim \sigma} [\underline{W}_p^p(\mu, \nu; P_\theta)],$$

where $\mathcal{M}(\mathbb{S}^{d-1}) \subset \mathcal{P}(\mathbb{S}^{d-1})$ be a set of probability measures on $\mathbb{S}^{d-1}$. When $\mathcal{M}(\mathbb{S}^{d-1}) = \{\delta_\theta \mid \theta \in \mathbb{S}^{d-1}\}$, max sliced Wasserstein distance (Deshpande et al., 2019) is obtained: Max-$SW(\mu, \nu) = \max_{\theta \in \mathbb{S}^{d-1}} \underline{W}_p(\mu, \nu; P_\theta)]$.

3. *Energy-based:* An optimization-free way to select $\sigma$ is to design it as an energy-based distribution with the unnormalized density: $p_\sigma(\theta) \propto f(\underline{W}_p^p(\mu, \nu; P_\theta))$, where $f$ is often chosen to be an increasing function on the positive real line, i.e., an exponential function. This choice of slicing distribution leads to energy-based SW (EBSW) (Nguyen & Ho, 2023).

**Empirical estimation.** For SW, Monte Carlo estimation is used to approximate the distance:

$$\widehat{SW}_p^p(\mu, \nu; \theta_1, \ldots, \theta_L) = \frac{1}{L} \sum_{l=1}^{L} \underline{W}_p^p(\mu, \nu; P_{\theta_l}),$$

where $\theta_1, \ldots, \theta_L \overset{i.i.d}{\sim} \mathcal{U}(\mathbb{S}^{d-1})$ $(L > 0)$ are projecting directions (other sampling techniques can also be used (Nguyen et al., 2024; Nguyen & Ho, 2024; Sisouk et al., 2025)). For Max-SW, we can use $\hat{\theta}_T$ which is the solution of an optimization algorithm with $T > 0$ iterations, e.g., projected gradient ascent (Nietert et al., 2022) or Riemannian gradient ascent Lin et al. (2020): $\widehat{\text{Max-SW}}_p^p(\mu, \nu; \hat{\theta}_T) = \underline{W}_p^p(\mu, \nu; P_{\hat{\theta}_T})$. For EBSW, one simple way to estimate the distance is to use importance sampling:

$$\widehat{EBSW}_p^p(\mu, \nu; \theta_1, \ldots, \theta_L) = \sum_{l=1}^{L} \hat{w}_l \underline{W}_p^p(\mu, \nu; P_{\theta_l}),$$

where $\hat{w}_l = \frac{f(W_p^p(\mu,\nu;P_{\theta_l}))}{\sum_{l'=1}^{L} f(W_p^p(\mu,\nu;P_{\theta_{l'}}))}$ and $\theta_1, \ldots, \theta_L \sim \mathcal{U}(\mathbb{S}^{d-1})$.

**Lower bounds.** We summarize the connection between SW, Max-SW, EBSW, and Wasserstein distance in the following remark. The detail of the proof can be found in Nguyen & Ho (2023).

**Remark 1.** *Given any $\mu \in \mathcal{P}_p(\mathbb{R}^d)$ and $\nu \in \mathcal{P}_p(\mathbb{R}^d)$, we have:*

*(a) $SW_p(\mu, \nu) \leq EBSW_p(\mu, \nu) \leq Max\text{-}SW_p(\mu, \nu) \leq W_p(\mu, \nu)$,*

*(b) $\widehat{SW}_p(\mu, \nu; \theta_1, \ldots, \theta_L) \leq \widehat{EBSW}_p(\mu, \nu; \theta_1, \ldots, \theta_L) \leq W_p(\mu, \nu)$ for any $\theta_1, \ldots, \theta_L \in \mathbb{S}^{d-1}$,*

*(c) $\widehat{Max\text{-}SW}_p^p(\mu, \nu; \hat{\theta}_T) \leq W_p(\mu, \nu)$ for any $\hat{\theta}_T \in \mathbb{S}^{d-1}$.*

**Lifted sliced Wasserstein distances.** Given $\mu \in \mathcal{P}_p(\mathbb{R}^d)$ and $\nu \in \mathcal{P}_p(\mathbb{R}^d)$, a lifted sliced Wasserstein-$p$ distance can be defined as follows (Rowland et al., 2019):

$$LSW_p^p(\mu, \nu; \sigma) = \mathbb{E}_{\theta \sim \sigma}\left[\overline{W}_p^p(\mu, \nu; P_\theta)\right], \tag{6}$$

where $P_\theta : \mathbb{R}^d \to \mathbb{R}$ is the projection function, $\overline{W}_p^p(\mu, \nu; P_\theta)$ is the SWGG (equation 3), and $\sigma \in \mathcal{P}(\mathbb{S}^{d-1})$ is the slicing distribution. Similar to SW, we can obtain variants of PW by choosing $\sigma$.

1. *Fixed prior:* The original LSW is introduced as in projected Wasserstein (PW) in Rowland et al. (2019), which uses the uniform distribution $\mathcal{U}(\mathbb{S}^{d-1})$.

2. *Optimization-based:* In contrast to the case of one-dimensional projected Wasserstein, which is always a lower bound of Wasserstein distance, SWGG is always an upper bound of Wasserstein distance. Therefore, it is desirable to select $\theta$ that can minimize the corresponding lifted cost, that leads to min SWGG distance:

$$\text{Min-}SWGG_p(\mu, \nu) = \min_{\theta \in \mathbb{S}^{d-1}} \overline{W}_p(\mu, \nu; P_\theta)$$

.

3. *Energy-based:* Similar to the case of EBSW, authors in Liu et al. (2025) proposes to choose $\sigma$ as an energy-based distribution with the unnormalized density: $p_\sigma(\theta) \propto f(-\underline{W}_p^p(\mu, \nu; P_\theta))$, where $f$ is often chosen to be an exponential function with temperature. The authors name the distance as expected sliced transport (EST).

**Empirical estimation.** For PW, Monte Carlo samples are used to approximate the distance: $\widehat{PW}_p^p(\mu, \nu; \theta_1, \ldots, \theta_L) = \frac{1}{L} \sum_{l=1}^{L} \overline{W}_p^p(\mu, \nu; P_{\theta_l})$, where $\theta_1, \ldots, \theta_L \overset{i.i.d}{\sim} \mathcal{U}(\mathbb{S}^{d-1})$. For Min-SWGG, we can use $\hat{\theta}_T$ which is the solution of an optimization algorithm with $T > 0$ iterations, e.g., simulated annealing (Mahey et al., 2023), gradient ascent with a surrogate objective (Mahey et al., 2023), and differentiable approximation (Chapel et al., 2025): $\widehat{\text{Min-}SWGG}_p^p(\mu, \nu; \hat{\theta}_T) = \overline{W}_p^p(\mu, \nu; P_{\hat{\theta}_T})$. For EST, importance sampling estimation is used:

$$\widehat{EST}_p^p(\mu, \nu; \theta_1, \ldots, \theta_L)) = \sum_{l=1}^{L} \hat{w}_l \overline{W}_p^p(\mu, \nu; P_{\theta_l}),$$

where $\hat{w}_l = \frac{f(-\overline{W}_p^p(\mu,\nu;P_{\theta_l}))}{\sum_{l'=1}^L f(-\overline{W}_p^p(\mu,\nu;P_{\theta_{l'}}))}$ and $\theta_1, \ldots, \theta_L \sim \mathcal{U}(\mathbb{S}^{d-1})$.

**Upper bounds.** We summarize the connection between PW, Min-SWGG, EST, and Wasserstein distance in the following remark. The connection between Min-SWGG, EST, and Wasserstein distance is discussed in Mahey et al. (2023); Liu et al. (2025). The connection between EST and PW can be generalized from the connection between EBSW and SW in Nguyen & Ho (2023).

**Remark 2.** *Given any $\mu \in \mathcal{P}_p(\mathbb{R}^d)$ and $\nu \in \mathcal{P}_p(\mathbb{R}^d)$, we have:*

*(a) $W_p(\mu,\nu) \leq Min\text{-}SWGG_p(\mu,\nu) \leq EST_p(\mu,\nu) \leq PW_p(\mu,\nu)$,*

*(b) $W_p(\mu,\nu) \leq \widehat{EST}_p(\mu,\nu;\theta_1,\ldots,\theta_L) \leq \widehat{PW}_p(\mu,\nu;\theta_1,\ldots,\theta_L)$ for any $\theta_1,\ldots,\theta_L \in \mathbb{S}^{d-1}$,*

*(c) $W_p(\mu,\nu) \leq Min\text{-}\widehat{SW}GG_p^p(\mu,\nu;\hat{\theta}_T)$ for any $\hat{\theta}_T \in \mathbb{S}^{d-1}$.*

## 3.2 REGRESSION OF WASSERSTEIN DISTANCE ON SLICED WASSERSTEIN DISTANCES

We consider the setting where we observe pairs of distributions $(\mu_1,\nu_1),\ldots,(\mu_N,\nu_N) \sim \mathbb{P}(\mu,\nu)$. Here, $\mathbb{P}(\mu,\nu)$ is the meta distribution, and we are interested in relating $W_p(\mu_i,\nu_i)$ with $K > 0$ SW distances $S_p^{(1)}(\mu_i,\nu_i),\ldots,S_p^{(K)}(\mu_i,\nu_i)$ for $i = 1,\ldots,N$. We first start with a general model.

**Definition 1** ( Regression of Wasserstein distance onto SW distances). *Given a meta distribution $\mathbb{P}(\mu,\nu) \in \mathcal{P}(\mathcal{P}_p(\mathbb{R}^d) \times \mathcal{P}_p(\mathbb{R}^d))$, $K > 0$ SW distances $S_p^{(1)},\ldots,S_p^{(K)}$, a regression model of Wasserstein distance onto SW distances is defined as follows:*

$$W_p(\mu,\nu) = f(S_p^{(1)}(\mu,\nu),\ldots,S_p^{(K)}(\mu,\nu)) + \varepsilon, \tag{7}$$

*where $(\mu,\nu) \sim \mathbb{P}(\mu,\nu)$, $f \in \mathcal{F}$ is the regression function, and $\varepsilon$ is a noise model such that $\mathbb{E}[\varepsilon] = 0$.*

To estimate $f$, one natural estimator is the least square estimate:

$$f_{LSE} = \arg\min_{f \in \mathcal{F}} \mathbb{E}\left[\left(f(S_p^{(1)}(\mu,\nu),\ldots,S_p^{(K)}(\mu,\nu)) - W_p(\mu,\nu)\right)^2\right]. \tag{8}$$

It is worth noting that the function $f$ can be constructed in both parametric ways (e.g., deep neural networks) or non-parametric ways (e.g., using kernels). However, in order to have a simple and explainable model, we consider linear functions in this work.

**Linear Regression of Wasserstein distance onto SW distances.** We now propose linear estimations of Wasserstein distances from SW distances.

**Definition 2** (Linear Regression of Wasserstein distance onto SW distances). *Given a meta distribution $\mathbb{P}(\mu,\nu) \in \mathcal{P}(\mathcal{P}_p(\mathbb{R}^d) \times \mathcal{P}_p(\mathbb{R}^d))$, $K > 0$ SW distances $S_p^{(1)},\ldots,S_p^{(K)}$, the linear regression model of Wasserstein distance onto SW distances is defined as follows:*

$$W_p(\mu,\nu) = \sum_{k=1}^K \omega_k S_p^{(k)}(\mu,\nu) + \varepsilon, \tag{9}$$

*where $(\mu,\nu) \sim \mathbb{P}(\mu,\nu)$ and $\varepsilon$ is a noise model such that $\mathbb{E}[\varepsilon] = 0$.*

Again, we use least-squares estimation to obtain an estimate of $\omega$.

**Remark 3.** *The least square estimator admits the following closed form:*

$$\boldsymbol{\omega}_{LSE} = \mathbb{E}\left[\boldsymbol{S}_p(\mu,\nu)\boldsymbol{S}_p(\mu,\nu)^\top\right]^{-1} \mathbb{E}\left[\boldsymbol{S}_p(\mu,\nu)W_p(\mu,\nu)\right], \tag{10}$$

*where $\boldsymbol{S}_p(\mu,\nu) = (S_p^{(1)}(\mu,\nu),\ldots,S_p^{(K)}(\mu,\nu))^\top$.*

The detail of Remark 3 in given in Appendix A.1. In practice, we can sample $(\mu_1,\nu_1),\ldots,(\mu_M,\nu_M) \sim \mathbb{P}(\mu,\nu)$ to approximate the expectation in equation 10. Let $\hat{\boldsymbol{S}} \in \mathbb{R}_+^{M \times K}$ be the SW distances matrix i.e., $\hat{\boldsymbol{S}}_{ik} = S_p^{(k)}(\mu_i,\nu_i)$ for $i = 1,\ldots,M$, and $\hat{\boldsymbol{W}} \in \mathbb{R}_+^M$ be the Wasserstein distances vector i.e., $\hat{\boldsymbol{W}}_i = W_p(\mu_i,\nu_i)$ for $i = 1,\ldots,M$, we have the sample-based least-squares

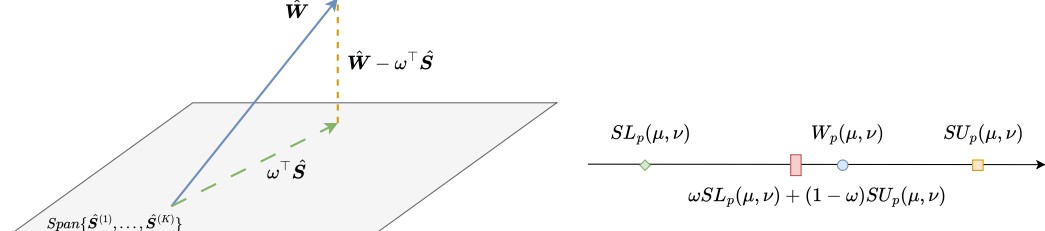

Figure 1: Linear regression of the Wasserstein distance vector $\hat{\boldsymbol{W}}$ on sliced Wasserstein (SW) distances $\hat{\boldsymbol{S}}^{(1)}, \ldots, \hat{\boldsymbol{S}}^{(K)}$. The left figure illustrates a linear model, interpreted as the $\mathbb{L}_2$ projection of the Wasserstein distance onto the linear span of the SW distances. The right figure depicts a special case of a constrained linear model with only two SW distances as predictors, which can be seen as a midpoint method.

estimate: $\hat{\boldsymbol{\omega}}_{LSE} = (\hat{\boldsymbol{S}}^{\top}\hat{\boldsymbol{S}})^{-1}\hat{\boldsymbol{S}}^{\top}\hat{\boldsymbol{W}}$, which is an unbiased estimate of $\boldsymbol{\omega}$. It is well-known that the linear model can be seen as $\mathbb{L}_2$ projection of the Wasserstein distances vector $\hat{\boldsymbol{W}}$ onto the linear span of the SW distances vectors $\hat{\boldsymbol{S}}^{(1)}, \ldots, \hat{\boldsymbol{S}}^{(K)}$. We illustrate the idea in the left figure in Figure 1.

From Section 3.1, we know that SW distances are either lower bounds or upper bounds of Wasserstein distance. Therefore, natural estimation can be formed using midpoint method. In particular, given a lower bound $SL_p(\mu, \nu)$ and a upper bound $SU_p(\mu, \nu)$, we can predict the Wasserstein distance as $\omega_1 SL_p(\mu, \nu) + \omega_2 SU_p(\mu, \nu)$ with $0 \le \omega_1 \le 1$ and $\omega_2 = 1 - \omega_1$.

**Definition 3** (Constrained Linear Regression of Wasserstein distance onto SW distances). *Given a meta distribution $\mathbb{P}(\mu, \nu) \in \mathcal{P}(\mathcal{P}_p(\mathbb{R}^d) \times \mathcal{P}_p(\mathbb{R}^d))$, $K > 0$ SW distances $SL_p^{(1)}, \ldots, SL_p^{(K)}$ which are lower bounds of $W_p$ and $K > 0$ SW distances $SU_p^{(1)}, \ldots, SU_p^{(K)}$ which are lower bounds of $W_p$, the constrained linear regression model is defined as follows:*

$$W_p(\mu, \nu) = \frac{1}{K} \sum_{k=1}^{K} \omega_k SL_p^{(k)}(\mu, \nu) + \frac{1}{K} \sum_{k=1}^{K} (1 - \omega_k) SU_p^{(k)}(\mu, \nu) + \varepsilon, \tag{11}$$

*where $0 \le \omega_k \le 1$, $(\mu, \nu) \sim \mathbb{P}(\mu, \nu)$ and $\varepsilon$ is a noise model such that $\mathbb{E}[\varepsilon] = 0$.*

To estimate $\boldsymbol{\omega} = (\omega_1, \ldots, \omega_K)$ under the constrained model, we again form the least square estimate, which can be solved using quadratic programming and Monte Carlo estimation. In a special case where $K = 1$, i.e., having one lower bound and one upper bound, we can have a closed-form.

**Remark 4.** *For the case $K = 1$ with a lower bound $SL_p(\mu, \nu)$ and an upper bound $SU_p(\mu, \nu)$, a closed-form of the least square estimate under the constrained model can be formed:*

$$\hat{\omega}_{CLSE} = \frac{\mathbb{E}\left[(SU_p(\mu, \nu) - SL_p(\mu, \nu))(SU_p(\mu, \nu) - W_p(\mu, \nu))\right]}{\mathbb{E}[(SU_p(\mu, \nu) - SL_p(\mu, \nu)^2]}. \tag{12}$$

The detail of Remark 4 in given in Appendix A.2. The corresponding sample-based estimator for the model is: $\hat{\omega}_{CLSE} = \frac{\frac{1}{M}\sum_{i=1}^{m}(SU_p(\mu_i, \nu_i) - SL_p(\mu_i, \nu_i))(SU_p(\mu_i, \nu_i) - W_p(\mu_i, \nu_i))}{\frac{1}{M}\sum_{i=1}^{M}(SU_p(\mu_i, \nu_i) - SL_p(\mu_i, \nu_i)^2}$. We show the idea in the right figure in Figure 1. Compared to the unconstrained model, the constrained model has half of the parameters. In addition, it adds inductive bias to the model, which is often helpful when having limited observed samples.

**Wasserstein Distance Estimation with Few-Shot Regression.** We recall that we observe $(\mu_1, \nu_1), \ldots, (\mu_N, \nu_N) \sim \mathbb{P}(\mu, \nu)$ in practice. It is not computationally efficient to compute the discussed least square estimates using all $N$ pairs of distributions since those estimates require evaluation of Wasserstein distances. We then sample a subset $(\mu_1', \nu_1'), \ldots, (\mu_N', \nu_M')$ from the original set with $M << N$. After obtaining an estimate $\hat{\boldsymbol{\omega}}$ from $(\mu_1', \nu_1'), \ldots, (\mu_N', \nu_M')$, we can form estimations of the Wasserstein distances for other pairs and any new pair of distributions given their SW distances.

**Computational complexities.** We assume that $N$ pairs of distributions have the number of supports be at most $n$ and in $d$ dimensions. For fitting the estimate on $M$ pairs, we need to compute $MK$ SW distances (using $L$ projecting directions) which costs $\mathcal{O}(MKLn(\log n + d))$ in time and $M$

Table 1: $k$-NN accuracy on point-cloud classification on ShapeNetV2 dataset.

| Methods | $R^2$ | $k$=1 | $k$=3 | $k$=5 | $k$=10 | $k$=15 |
|---|---|---|---|---|---|---|
| WD | – | 83.6% ± 0.0% | 83.5% ± 0.0% | 84.2% ± 0.0% | 82.9% ± 0.0% | 79.2% ± 0.0% |
| RG-s | 0.868 ± 0.02 | 82.1% ± 0.1% | 81.7% ± 0.1% | 80.8% ± 0.1% | 79.4% ± 0.2% | 75.5% ± 0.2% |
| RG-e | 0.926 ± 0.04 | 82.5% ± 0.1% | 82.2% ± 0.1% | 80.9% ± 0.2% | 79.6% ± 0.3% | 75.7% ± 0.3% |
| RG-o | 0.774 ± 0.38 | 65.1% ± 0.3% | 67.7% ± 0.3% | 67.6% ± 0.5% | 66.7% ± 0.5% | 66.0% ± 0.5% |
| RG-se | 0.935 ± 0.02 | 82.5% ± 0.4% | 82.2% ± 0.4% | 82.6% ± 0.5% | 81.9% ± 0.5% | 76.5% ± 0.5% |
| RG-seo | 0.937 ± 0.01 | 82.8% ± 0.4% | 83.3% ± 0.5% | 83.5% ± 0.7% | 82.3% ± 0.7% | 77.9% ± 0.7% |

Wasserstein distances which costs $\mathcal{O}(Mn^2(n \log n + d))$. Computing the least square estimate has the time complexity of $\mathcal{O}(MK^2 + K^3)$. Then, we compute $(N - M)K$ SW distances which costs $\mathcal{O}((N-M)KLn(\log n + d))$ and predict $(N-M)$ Wasserstein distances which costs $\mathcal{O}((N-M)K)$. Total time complexity is $\mathcal{O}(NKLn(\log n + d)) + Mn^2(n \log n + d) + MK^2 + K^3 + (N - M)K)$ compared to $\mathcal{O}(Nn^2(n \log n + d))$ of computing Wasserstein distances for all $N$ pairs.

**Extensions on regression.** In this work, we focus on regressing the Wasserstein-$p$ distance. If other ground metrics are used e.g., geodesic distances on manifolds, variants of SW distances such as spherical sliced Wasserstein distances (Bonet et al., 2023a; Tran et al., 2024; Quellmalz et al., 2023), hyperbolic sliced Wasserstein distances (Bonet et al., 2023b), sliced Wasserstein for distributions over positive definite matrices (Bonet et al., 2023c), and other non-linear variants of sliced Wasserstein (Bonet et al., 2025; Chapel et al., 2025; Tanguy et al., 2025; Kolouri et al., 2019). However, they might not be upper/lower bounds of the corresponding Wasserstein distances. Moreover, to incorporate uncertainty quantification, we can also perform Bayesian inference (Box & Tiao, 2011), e.g., putting a prior on the regression function.

## 4 EXPERIMENTS

We define some specific model instances: *RG-o* uses Max-SW and Min-SWGG as predictors; *RG-s* uses SW and PWD as predictors; *RG-e* uses EBSW and EST as predictors. We also consider two extensions: *RG-se* combines SW, EBSW, PWD, and EST, and *RG-seo* combines all six variants. For each instance, we have a *constrained* version and an *unconstrained* version as discussed.

We evaluate our methods in several parts, each with a distinct goal. First, in Section 4.1, we test practical use via $k$-NN on ShapeNetV2, reporting accuracy under different metrics. Second, in Section 4.2, we benchmark *RG* variants against Wormhole across MNIST point clouds, ShapeNetV2, MERFISH Cell Niches Zhang et al. (2021), and scRNA-seq atlas Persad et al. (2023), reporting $R^2$/MSE/MAE in low-data regimes. Third, in Section 4.3, we combine our framework with Wormhole to introduce *RG-Wormhole*, a hybrid that matches Wormhole's performance while requiring far less training time. We compare training time under varying batch sizes and epochs, as well as embedding, reconstruction, barycenter, and interpolation quality. In Appendix B.1, we run MoG simulations to verify that our methods approximate the true Wasserstein from low to high dimensions. In Appendix B.3, we visualize metric-induced geometry with UMAP McInnes et al. (2018). In Appendix B.6, we compare *RG* with classical methods in the many-pairs setting. To illustrate robustness, we further evaluate the method under inter-class and intra-class settings in Appendix B.7. Finally, we investigate whether there are consistent patterns in the optimal *RG* weights across datasets; see Appendix B.8. Throughout, $N$ denotes the number of training-set sizes, and $M_0$ the number of samples drawn from the training set, yielding $M = \frac{M_0(M_0-1)}{2}$ pairs used to estimate *RG* coefficients.

### 4.1 POINT CLOUD CLASSIFICATION

We evaluate unconstrained *RG* variants over a classification task over 10-class ShapeNetV2 with 500 training samples ($N$=500) and estimate *RG* weights from 10 samples ($M_0$=10) drawn from the training set. The details of the experimental setting and full results are provided in Appendix B.2.

**Results.** Table 1 reports $k$-NN accuracy on ShapeNetV2 under different metrics. As expected, WD achieves the best accuracy, with $84.2\%$ at $k$=5. Among single sliced-based metrics, SW and EBSW, are the strongest, though they cap at about $72.5\%$ top-1. Our *RG* methods close much of the gap to Wasserstein. Both *RG-s* and *RG-e* consistently achieve around $82.5\%$ top-1 accuracy with high

Table 2: Approximation quality of Wormhole and $RG$ variants across four datasets under a training set size of 100 samples. Each cell reports $R^2$, MSE, and MA) with respect to the exact Wasserstein distance.

| Methods | MNIST Point Cloud | | | ShapeNetV2 | | | MERFISH | | | scRNA-seq | | |
|---|---|---|---|---|---|---|---|---|---|---|---|---|
| | $R^2$ | MSE | MAE | $R^2$ | MSE | MAE | $R^2$ | MSE | MAE | $R^2$ | MSE | MAE |
| Wormhole | 0.28 | $4.3 \times 10^{-1}$ | $5.1 \times 10^{-1}$ | 0.65 | $6.6 \times 10^{-2}$ | $1.8 \times 10^{-1}$ | -3.6 | $8.0 \times 10^{-4}$ | $2.1 \times 10^{-2}$ | 0.04 | $7.0 \times 10^{-3}$ | $7.8 \times 10^{-2}$ |
| RG-s (constr.) | 0.84 | $8.9 \times 10^{-2}$ | $2.3 \times 10^{-1}$ | 0.88 | $2.0 \times 10^{-2}$ | $1.1 \times 10^{-1}$ | 0.91 | $1.6 \times 10^{-5}$ | $3.0 \times 10^{-3}$ | 1.00 | $3.7 \times 10^{-5}$ | $3.0 \times 10^{-3}$ |
| RG-e (constr.) | 0.86 | $8.7 \times 10^{-2}$ | $2.3 \times 10^{-1}$ | 0.90 | $1.7 \times 10^{-2}$ | $1.0 \times 10^{-1}$ | 0.92 | $1.3 \times 10^{-5}$ | $3.0 \times 10^{-3}$ | 1.00 | $1.3 \times 10^{-5}$ | $1.0 \times 10^{-3}$ |
| RG-o (constr.) | 0.77 | $1.4 \times 10^{-1}$ | $2.8 \times 10^{-1}$ | 0.66 | $5.2 \times 10^{-2}$ | $1.8 \times 10^{-1}$ | 0.75 | $4.8 \times 10^{-5}$ | $6.0 \times 10^{-3}$ | 0.99 | $6.1 \times 10^{-5}$ | $6.0 \times 10^{-3}$ |
| RG-se (constr.) | 0.84 | $9.8 \times 10^{-2}$ | $2.4 \times 10^{-1}$ | 0.92 | $1.4 \times 10^{-2}$ | $9.3 \times 10^{-2}$ | 0.91 | $1.5 \times 10^{-5}$ | $3.0 \times 10^{-3}$ | 1.00 | $2.4 \times 10^{-5}$ | $2.0 \times 10^{-3}$ |
| RG-seo (constr.) | 0.85 | $9.0 \times 10^{-2}$ | $2.3 \times 10^{-1}$ | 0.91 | $1.7 \times 10^{-2}$ | $1.0 \times 10^{-1}$ | 0.92 | $1.3 \times 10^{-5}$ | $3.0 \times 10^{-3}$ | 1.00 | $2.2 \times 10^{-5}$ | $2.0 \times 10^{-3}$ |
| RG-s (unconstr.) | 0.93 | $4.5 \times 10^{-2}$ | $1.6 \times 10^{-1}$ | 0.94 | $1.1 \times 10^{-2}$ | $8.2 \times 10^{-2}$ | 0.96 | $6.3 \times 10^{-6}$ | $2.0 \times 10^{-3}$ | 0.99 | $8.6 \times 10^{-5}$ | $7.0 \times 10^{-3}$ |
| RG-e (unconstr.) | 0.92 | $5.4 \times 10^{-2}$ | $1.8 \times 10^{-1}$ | 0.92 | $1.5 \times 10^{-2}$ | $9.8 \times 10^{-2}$ | 0.96 | $6.9 \times 10^{-6}$ | $2.0 \times 10^{-3}$ | 0.99 | $7.0 \times 10^{-5}$ | $6.0 \times 10^{-3}$ |
| RG-o (unconstr.) | 0.77 | $1.4 \times 10^{-1}$ | $3.0 \times 10^{-1}$ | 0.75 | $3.8 \times 10^{-2}$ | $1.6 \times 10^{-1}$ | 0.89 | $8.7 \times 10^{-4}$ | $2.9 \times 10^{-2}$ | 0.82 | $2.9 \times 10^{-3}$ | $5.2 \times 10^{-2}$ |
| RG-se (unconstr.) | 0.93 | $4.0 \times 10^{-2}$ | $1.5 \times 10^{-1}$ | 0.95 | $9.9 \times 10^{-3}$ | $7.8 \times 10^{-2}$ | 0.98 | $2.9 \times 10^{-6}$ | $1.0 \times 10^{-3}$ | 1.00 | $3.0 \times 10^{-5}$ | $4.0 \times 10^{-3}$ |
| RG-seo (unconstr.) | 0.93 | $4.0 \times 10^{-2}$ | $1.5 \times 10^{-1}$ | 0.95 | $9.8 \times 10^{-3}$ | $7.8 \times 10^{-2}$ | 0.97 | $6.8 \times 10^{-6}$ | $2.0 \times 10^{-3}$ | 0.99 | $6.8 \times 10^{-5}$ | $7.0 \times 10^{-3}$ |

Figure 2: ModelNet40: a *RG-Wormhole* variant in reconstruction experiment.

correlation to Wasserstein ($R^2 \approx 0.9$). The multi-metric extensions further improve stability: *RG-se* and *RG-seo* reach up to $83.5\%$ accuracy with $R^2$ as high as $0.93$, essentially matching Wasserstein.

## 4.2 COMPARISONS OF RG VARIANTS VS. WORMHOLE IN LOW-DATA REGIMES

We compare our *RG* framework with Wormhole within the same training sizes, matching the pre-processing of (Haviv et al., 2024) across four datasets spanning dimensionality: MNIST pixel point clouds (2D), ShapeNetV2 point clouds (3D), MERFISH Niche Cells (254D), and scRNA-seq (2,500D). We train on $N \in \{10, 50, 100, 200\}$ random pairs and evaluate $R^2$/MSE/MAE against exact WD. For fairness, the number of training pairs for Wormhole equals the number used to estimate the linear coefficients for *RG* variants, i.e., $M_0 = N$. Full results appear in Figures 6–13 with settings in Appendix B.4; Table 2 summarizes the $M_0 = 100$ case, and other $M_0$ follow the same pattern.

**Results.** Across all four datasets, *RG* variants consistently outperform Wormhole at small training sizes. Wormhole is weaker primarily because it is data hungry and its performance improves as we add samples, yet under comparable budgets it still trails our methods. By contrast, *RG* variants are already accurate with few pairs, with *unconstrained* variants are slightly stronger, whereas *constrained* variants converge faster and are preferable at the very smallest sizes. *RG-se* and *RG-seo* are the strongest when given sufficient samples though the latter can lag at the tiniest sizes before its weights settle but becomes top-performing quickly and still requires far fewer pairs than Wormhole.

## 4.3 RG-WORMHOLE: ACCELERATING WORMHOLE WITH REGRESSION OF WASSERSTEIN

The previous comparison reveals a clear trade-off. *RG* framework is lightweight and data-efficient, but it does not produce Euclidean embeddings and therefore cannot support interpolation experiments. Wormhole, in contrast, learns Euclidean embeddings that enable interpolation and reconstruction, but it is computationally heavy because training requires many Wasserstein evaluations (pairwise distances within each mini-batch and reconstruction losses), which slows and raises training cost.

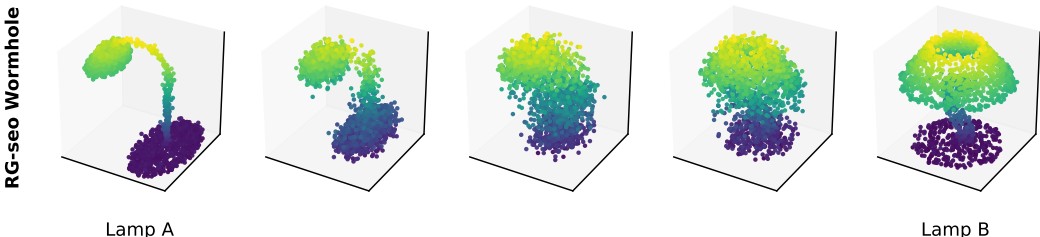

Figure 3: ModelNet40: a *RG-Wormhole* variant in interpolation experiment.

**RG-Wormhole.** To combine the strengths of both, we introduce *RG-Wormhole*. We first calibrate a *RG* surrogate on a small set of exact Wasserstein pairs from the same data domain and freeze its weights. We then keep the Wormhole architecture, optimizer, and schedule unchanged, and simply replace every use of the Wasserstein distance with the calibrated surrogate in both the encoder (pairwise distances in the batch) and the decoder (reconstruction loss). No other component is modified. This substitution makes each training step much faster while preserving the performance.

We run five experiments of both models to empirically show that *RG-Wormhole* is much faster than Wormhole while keeping similar effectiveness. First, we measure training time by training Wormhole and *RG-Wormhole* under the same optimizer and schedule, sweeping batch sizes 4–20 and reporting wall-clock time for training-set sizes $N \in \{10, 50, 100, 200\}$. Second, we assess encoders via $R^2$/MSE/MAE between learned pairwise distances and exact Wasserstein. Third, we evaluate decoders via the Wasserstein loss between each input shape and its reconstruction. Fourth, we examine barycenters by decoding each class's mean embedding and visualizing results. Finally, we study interpolation by decoding linear paths between two embeddings and visualizing the trajectories. Across all experiments, hyperparameters match Wormhole; the only change in *RG-Wormhole* is replacing every use of the Wasserstein distance in the encoder and decoder losses with the calibrated unconstrained *RG* variants. For *RG-Wormhole*, we estimate the *RG* coefficient using 10 random training samples ($M_0{=}10$) before plugging into Wormhole. We provide some results in Figures 2–3 though the details of experimental settings and full results can be found in Appendix B.5.

**Results.** Replacing every Wasserstein call in Wormhole with a calibrated *RG* variants preserves performance while cutting compute. First, in the training-time comparison (Figure 14 in Appendix B.5), *RG-Wormhole* is far faster than Wormhole across all batch sizes and training budgets, with a very large gap. As batch size increases, Wormhole's time grows almost exponentially, while *RG-Wormhole* rises only slightly, close to linear or even flat. Next, we verify that the trained models have similar quality. For the encoder, Figures 15 and 16 in Appendix B.5 show pairwise distances that align with the ground-truth Wasserstein and embeddings that match Wormhole, with essentially identical $R^2$, MSE, and MAE. For the decoder, Figures 17 and 18 in Appendix B.5 evaluate reconstructions against the original point clouds using the Wasserstein distance, and both *RG-Wormhole* and Wormhole produce very small and nearly identical distances. Finally we test whether *RG-Wormhole* preserves the geometry needed for downstream use. The decoded class barycenters from *RG-Wormhole* are clean and class consistent and they match those from Wormhole, we refer to Figure 19 in Appendix B.5. We also interpolate by moving linearly in the embedding space and decoding along the path, and the trajectories from *RG-Wormhole* are smooth and semantically meaningful with no visible artifacts, we refer to Figure 20 in Appendix B.5. Overall *RG-Wormhole* matches Wormhole while training much faster, which makes it a practical choice when compute is limited.

## 5 CONCLUSIONS

We introduced a regression framework mapping Wasserstein to sliced Wasserstein distances under a meta-distribution of random distribution pairs. Two simple linear models enable lightweight estimation, leading to the *RG* framework for few-shot Wasserstein approximation. We derived constrained and unconstrained forms and validated them through Mixture of Gaussian simulations, point cloud classification, and metric-space visualizations, where the surrogate closely matched the exact distance. Compared to Wormhole on MNIST, ShapeNetV2, MERFISH, and scRNA-seq, our method achieved better performance in low-data regimes. Replacing Wasserstein calls in Wormhole with our method yielded *RG-Wormhole*, preserving accuracy while greatly reducing training time.

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

# Supplement to "Fast Estimation of Wasserstein Distances via Regression on Sliced Wasserstein Distances"

## A DETAILS

### A.1 DETAILS OF REMARK 3

We derive the gradient:

$$
\nabla_{\boldsymbol{\omega}} \mathbb{E} \left[ \left\| \boldsymbol{\omega}^{\top} \boldsymbol{S}_p^{(k)}(\mu, \nu) - W_p(\mu, \nu)) \right\|_2^2 \right]
$$

$$
= \nabla_{\boldsymbol{\omega}} \mathbb{E} \left[ (\boldsymbol{\omega}^{\top} \boldsymbol{S}_p^{(k)}(\mu, \nu) - W_p(\mu, \nu)))^{\top} (\boldsymbol{\omega}^{\top} \boldsymbol{S}_p^{(k)}(\mu, \nu) - W_p(\mu, \nu))) \right]
$$

$$
= \nabla_{\boldsymbol{\omega}} \mathbb{E} \left[ \boldsymbol{\omega}^{\top} \boldsymbol{S}_p^{(k)}(\mu, \nu) \boldsymbol{S}_p^{(k)}(\mu, \nu)^{\top} \boldsymbol{\omega} \right] - 2 \nabla_{\boldsymbol{\omega}} \mathbb{E} \left[ \boldsymbol{S}_p^{(k)}(\mu, \nu)^{\top} \boldsymbol{\omega} W_p(\mu, \nu) \right] \tag{13}
$$

$$
= \mathbb{E} \left[ \nabla_{\boldsymbol{\omega}} \boldsymbol{\omega}^{\top} \boldsymbol{S}_p^{(k)}(\mu, \nu) \boldsymbol{S}_p^{(k)}(\mu, \nu)^{\top} \boldsymbol{\omega} \right] - 2 \mathbb{E} \left[ \nabla_{\boldsymbol{\omega}} \boldsymbol{S}_p^{(k)}(\mu, \nu)^{\top} \boldsymbol{\omega} W_p(\mu, \nu) \right] \tag{14}
$$

$$
= 2 \mathbb{E} \left[ \boldsymbol{S}_p^{(k)}(\mu, \nu) \boldsymbol{S}_p^{(k)}(\mu, \nu)^{\top} \right] \boldsymbol{\omega} - 2 \mathbb{E} \left[ \boldsymbol{S}_p^{(k)}(\mu, \nu) W_p(\mu, \nu) \right] \tag{15}
$$

Setting the gradient to 0, we obtain

$$
\hat{\boldsymbol{\omega}}_{LSE} = \mathbb{E} \left[ \boldsymbol{S}_p^{(k)}(\mu, \nu) \boldsymbol{S}_p^{(k)}(\mu, \nu)^{\top} \right]^{-1} \mathbb{E} \left[ \boldsymbol{S}_p^{(k)}(\mu, \nu) W_p(\mu, \nu) \right], \tag{16}
$$

which completes the proof.

### A.2 DETAILS OF REMARK 4

From the definition, we recall the model:

$$
W_p(\mu, \nu) = \sum_{k=1}^{K} \omega_k SL_p^{(k)}(\mu, \nu) + \sum_{k=1}^{K} (1 - \omega_k) SU_p^{(k)}(\mu, \nu) + \varepsilon. \tag{17}
$$

With $K = 1$, we rewrite the model as follows:

$$
W_p(\mu, \nu) = \omega SL_p(\mu, \nu) + (1 - \omega) SU_p(\mu, \nu) + \varepsilon, \tag{18}
$$

which is equivalent to

$$
W_p(\mu, \nu) - SU_p(\mu, \nu) = \omega (SL_p(\mu, \nu) - SU_p(\mu, \nu)) + \epsilon. \tag{19}
$$

Since equation 19 is again an unconstrained linear model, we can obtain the least-squares estimate by following Appendix A.1:

$$
\hat{\omega}_{CLSE} = \frac{\mathbb{E} \left[ (SU_p(\mu, \nu) - SL_p(\mu, \nu))(SU_p(\mu, \nu) - W_p(\mu, \nu)) \right]}{\mathbb{E} [(SU_p(\mu, \nu) - SL_p(\mu, \nu)^2]}, \tag{20}
$$

which concludes the proof.

## B EXPERIMENTS

### B.1 GAUSSIAN SIMULATION

We study how a lower–upper bound pair approximates the Wasserstein distance as dimension grows. We simulate 3-component Gaussian mixtures for $d=1 \ldots 100$ (10 seeds), with 200 points per component. For each pair we compute the exact Wasserstein and six sliced-based metrics. Focusing on *RG-o*, *RG-s*, and *RG-e*, we fit a constrained weight $w \in [0, 1]$ and report the estimated weight $\hat{w}$ and $R^2$ versus the exact Wasserstein.

**Results.** We refer to Figure 4 for the result. The fits are strong for all three methods and all dimensions: $R^2$ is always above 0.8 and quickly rises to $\approx$ 0.9-1.0. We also see a clear pattern in the weights: as dimension grows, the weight on the lower bound goes down, so the upper-bound metric gets more weight and eventually dominates. In short, high dimensions favor the upper bound, while lower dimensions rely more on the lower bound.

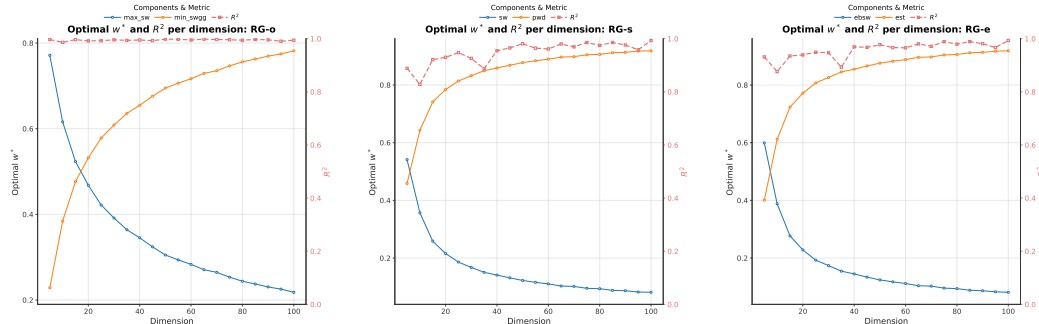

Figure 4: Optimal $w^*$ and $R^2$ in each dimension.

Table 3: $k$-NN accuracy on point-cloud classification on ShapeNetV2 dataset.

| Methods | $R^2$ | $k{=}1$ | $k{=}3$ | $k{=}5$ | $k{=}10$ | $k{=}15$ |
|---|---|---|---|---|---|---|
| WD | – | $83.6\% \pm 0.0\%$ | $83.5\% \pm 0.0\%$ | $84.2\% \pm 0.0\%$ | $82.9\% \pm 0.0\%$ | $79.2\% \pm 0.0\%$ |
| SWD | – | $72.4\% \pm 0.0\%$ | $71.4\% \pm 0.0\%$ | $70.4\% \pm 0.0\%$ | $69.0\% \pm 0.0\%$ | $66.7\% \pm 0.0\%$ |
| PWD | – | $42.6\% \pm 0.0\%$ | $42.9\% \pm 0.0\%$ | $40.4\% \pm 0.0\%$ | $39.3\% \pm 0.0\%$ | $39.0\% \pm 0.0\%$ |
| EBSW | – | $72.5\% \pm 0.0\%$ | $69.2\% \pm 0.0\%$ | $60.4\% \pm 0.0\%$ | $67.9\% \pm 0.0\%$ | $65.3\% \pm 0.0\%$ |
| EST | – | $39.1\% \pm 0.0\%$ | $40.4\% \pm 0.0\%$ | $40.2\% \pm 0.0\%$ | $38.0\% \pm 0.0\%$ | $36.5\% \pm 0.0\%$ |
| Max-SW | – | $60.3\% \pm 0.0\%$ | $54.6\% \pm 0.0\%$ | $57.7\% \pm 0.0\%$ | $57.6\% \pm 0.0\%$ | $56.8\% \pm 0.0\%$ |
| Min-SWGG | – | $36.4\% \pm 0.0\%$ | $37.6\% \pm 0.0\%$ | $35.0\% \pm 0.0\%$ | $32.9\% \pm 0.0\%$ | $30.8\% \pm 0.0\%$ |
| RG-s | $0.868 \pm 0.02$ | $82.1\% \pm 0.1\%$ | $81.7\% \pm 0.1\%$ | $80.8\% \pm 0.1\%$ | $79.4\% \pm 0.2\%$ | $75.5\% \pm 0.2\%$ |
| RG-e | $0.926 \pm 0.04$ | $82.5\% \pm 0.1\%$ | $82.2\% \pm 0.1\%$ | $80.9\% \pm 0.2\%$ | $79.6\% \pm 0.3\%$ | $75.7\% \pm 0.3\%$ |
| RG-o | $0.774 \pm 0.38$ | $65.1\% \pm 0.3\%$ | $67.7\% \pm 0.3\%$ | $67.6\% \pm 0.5\%$ | $66.7\% \pm 0.5\%$ | $66.0\% \pm 0.5\%$ |
| RG-se | $0.935 \pm 0.02$ | $82.5\% \pm 0.4\%$ | $82.2\% \pm 0.4\%$ | $82.6\% \pm 0.5\%$ | $81.9\% \pm 0.5\%$ | $76.5\% \pm 0.5\%$ |
| RG-seo | $0.937 \pm 0.01$ | $82.8\% \pm 0.4\%$ | $83.3\% \pm 0.5\%$ | $83.5\% \pm 0.7\%$ | $82.3\% \pm 0.7\%$ | $77.9\% \pm 0.7\%$ |

## B.2 POINT CLOUD CLASSIFICATION

**Experimental settings.** We construct a 10-class subset, centralize, normalize each shape so that all coordinates lie in $[-1, 1]^3$, and uniformly subsample $2{,}048$ points per shape. For each class we select 50 training examples and 100 test examples. We then compute pairwise distance matrices between train and test sets under different metrics, and evaluate classification accuracy using a $k$-nearest neighbor classifier with $k \in \{1, 3, 5, 10, 15\}$. Besides the six individual sliced-based metrics, we include all $RG$ variants in unconstrained version. We use 10 samples drawn from the training set to estimate the linear coefficient of $RG$ variants.

## B.3 METRIC SPACE VISUALIZATION

**Experimental settings.** We visualize the geometry each metric induces on ShapeNetV2. From 10 categories, we randomly sample 500 shapes per class, normalize each shape so that all coordinates lie in $[-1, 1]^3$, and keep $2{,}048$ points per shape. For every method, we compute the pairwise distance matrix, then feed to UMAP to obtain 2D embeddings. We use 10 samples drawn from the training set to estimate the linear coefficient of $RG$ variants.

**Results.** The result is visual in Figures 5. Across methods, the true Wasserstein produces well-separated class clusters with clear margins. The $RG$ variants produce embeddings that are visually very close to the Wasserstein embeddings, preserving both local compactness and the global arrangement of classes. By contrast, single sliced baselines are weaker. SWD and EBSW keep some structure but blur boundaries, while Max-SW and Min-SWGG show more mixing and noise.

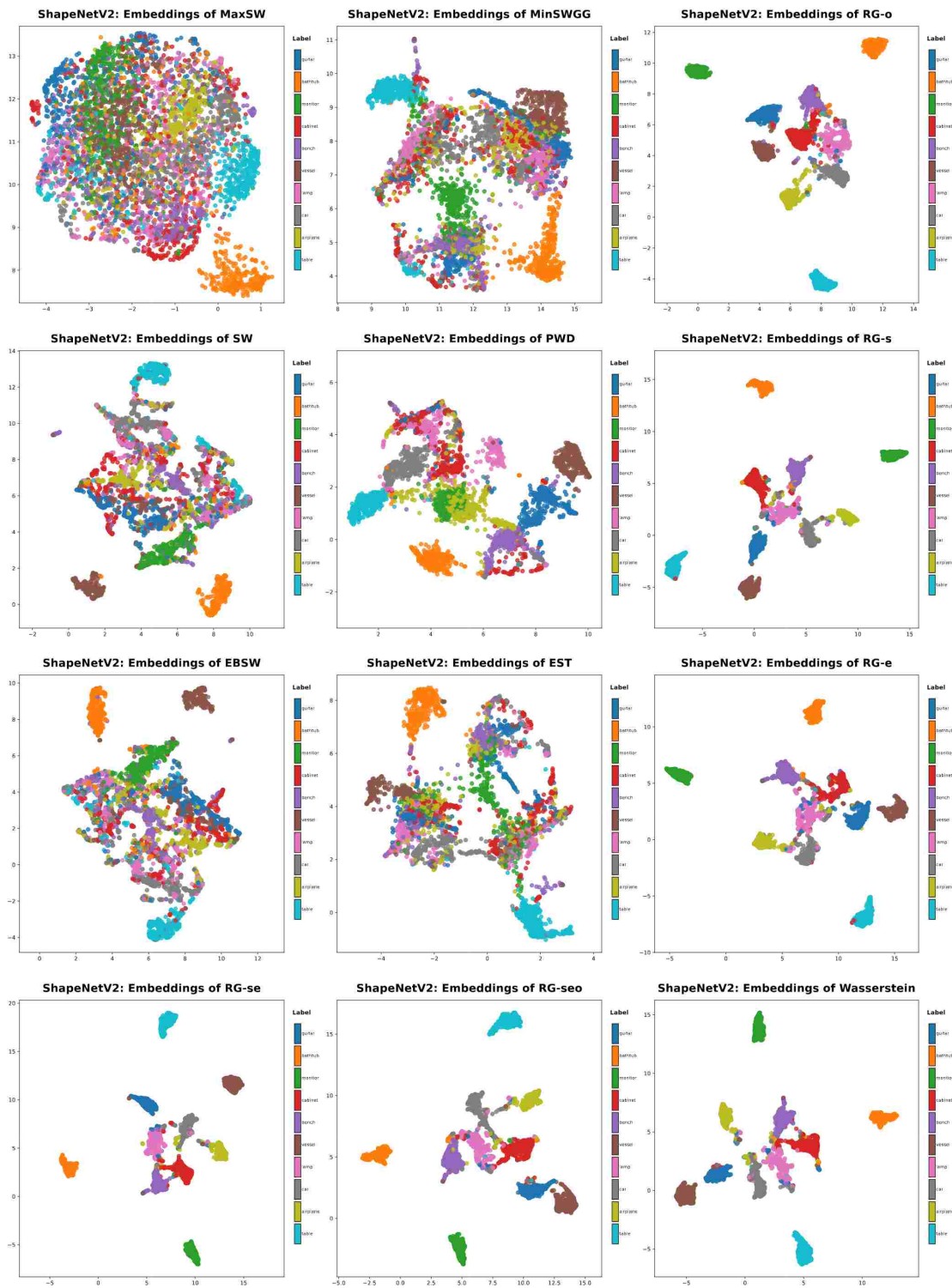

Figure 5: Embeddings of methods in ShapeNetV2 dataset.

Table 4: Wormhole training hyperparameters.

| Component | Setting |
|---|---|
| Batch size | `10` |
| Optimizer / LR | Adam, `lr` $= 10^{-4}$ |
| LR schedule | ExponentialLR, final factor $\approx 0.1$ over all epochs |
| Epochs | 2,000 epochs (20,000 steps) |
| Transformer depth | `num_layers` $= 3$ |
| Attention heads | `num_heads` $= 4$ |
| Embedding dim | `emb_dim` $= 128$ |
| MLP hidden dim | `mlp_dim` $= 512$ |
| Attention dropout | `attention_dropout_rate` $= 0.1$ |
| Decoder coeff. | `coeff_dec` $= 0.1$ |

### B.4 COMPARISON OF RG VARIANTS VS. WORMHOLE IN LOW-DATA REGIMES

**Experimental Settings.** We compare our proposed *RG* framework against Wormhole, a state-of-the-art Wasserstein approximation method. To ensure fairness, we follow the exact preprocessing protocol of Haviv et al. (2024). We consider four datasets spanning a wide range of dimensionalities: (i) MNIST point clouds, obtained by thresholding $28 \times 28$ grayscale images and treating the active pixels as 2D point coordinates; (ii) ShapeNetV2 point clouds, where each CAD model is uniformly sampled into $2,048$ points in 3D and normalized; (iii) MERFISH Cell Niches, where each cell is represented by the $50\,\mu$m neighborhood of its gene-expression profile embedded in a 254-dimensional space; and (iv) scRNA-seq atlas data, where cells are aggregated into MetaCells that form 2,500-dimensional gene-expression point clouds. We vary the number of training pairs $N \in \{10, 50, 100, 200\}$ by drawing pairs uniformly, and evaluate on 10,000 independently sampled test pairs. For each dataset and training size, we report $R^2$, MSE, and MAE with respect to the exact Wasserstein.

The original Wormhole codebase is built on JAX and TensorFlow, which are not compatible with our environment. Accordingly, we reimplemented Wormhole in PyTorch.

**Data Preprocessing.** We follow the same preprocessing pipeline as Haviv et al. (2024).

- **MNIST Point Clouds.** We turn MNIST $28 \times 28$ images into 2D point clouds by thresholding pixel values at $0.5$ and keeping the coordinates of the active pixels.

- **ShapeNetV2 Point Clouds.** We use ShapeNetCore.v2 with 15k points per shape. Each shape is normalized to fit inside a unit cube with coordinates in $[-1, 1]^3$. We then split each shape into 10k training points and 5k test points, and randomly sample $2,048$ points from each point cloud.

- **MERFISH Cell Niches.** We scale each gene's expression to $[-1, 1]$ and divide by $\sqrt{d}$, where $d$ is the number of genes. For each cell, we use spatial positions to find its 11 nearest neighbors within a $50\,\mu$m radius, keeping only cells with enough neighbors with its cell-type label.

- **scRNA-seq.** We select $2,500$ highly variable genes, normalize counts (library-size $10^4$ and $\log(1+x)$), and scale each gene to $[-1, 1]$ divided by $\sqrt{d}$ ($d{=}2500$). We then cluster cells with $K$-means. For each cluster seed, we consider it as a cloud, labeled by the seed's annotation.

**Wormhole training hyperparameters.** We follow the Transformer autoencoder setup of *Wormhole* with the configuration in Table 4.

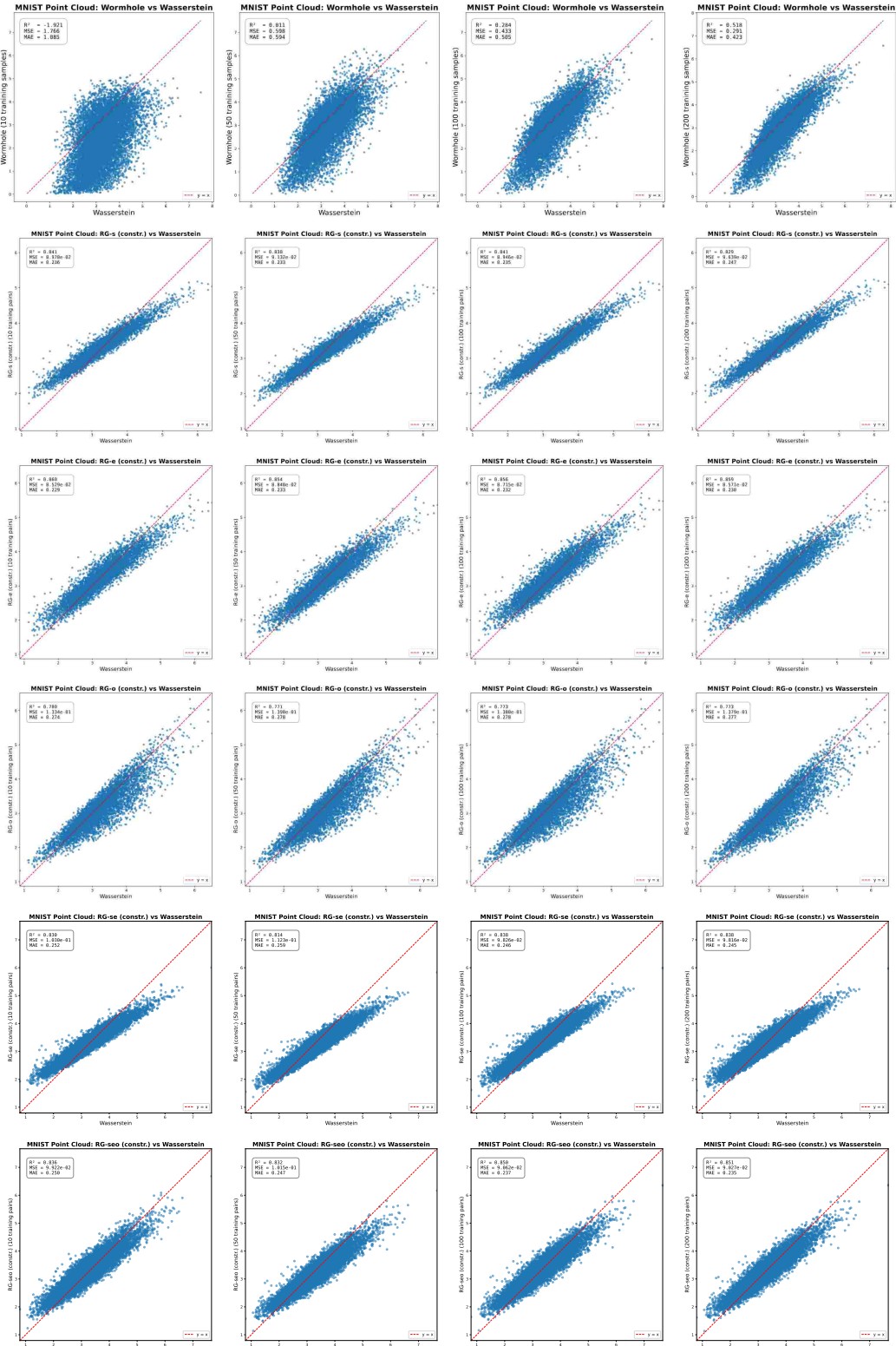

Figure 6: MNIST Point Cloud: Wormhole and *RG* variants (constrained/unconstrained) across training set sizes of 10, 50, 100 and 200.

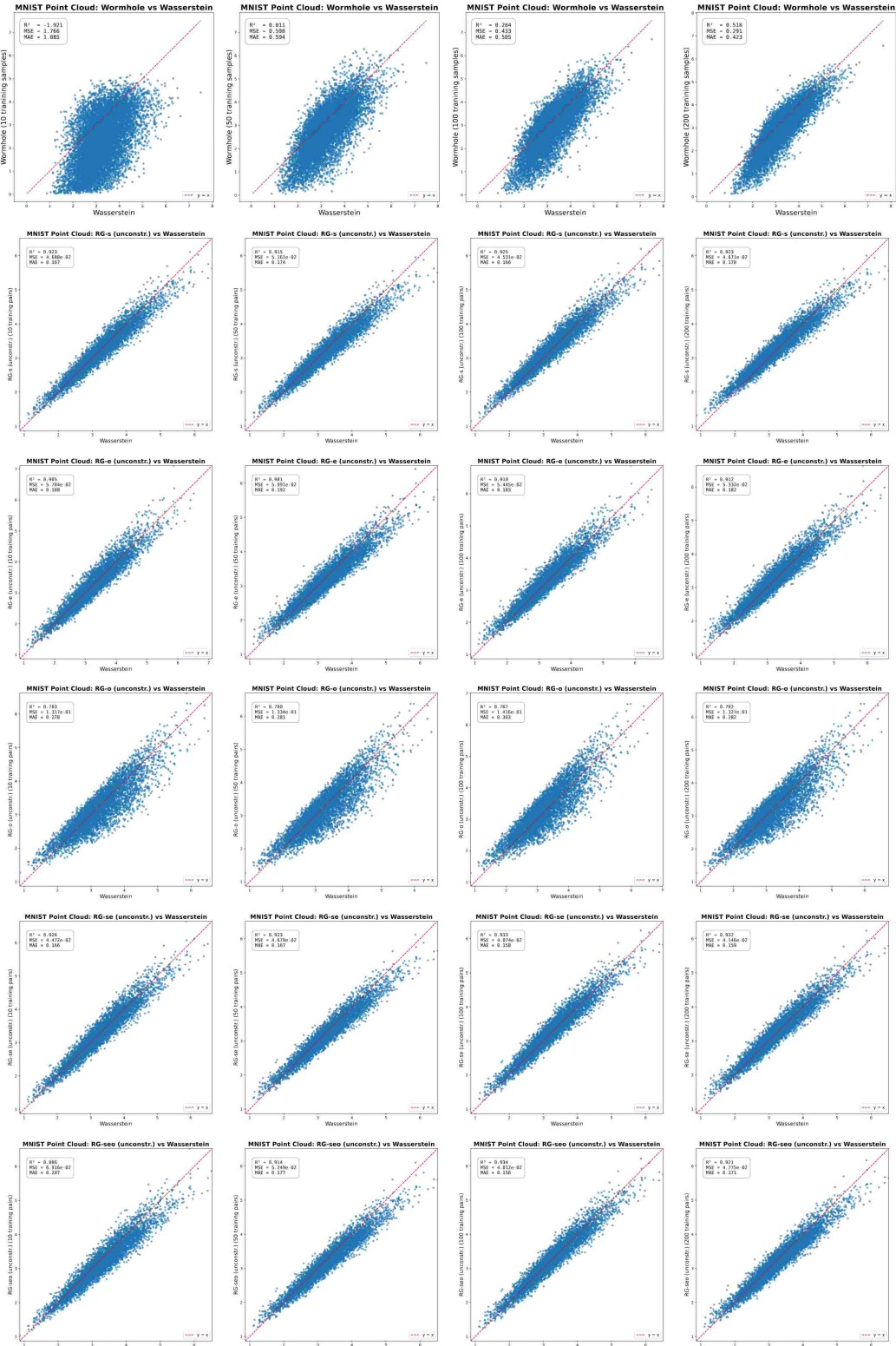

Figure 7: MNIST Point Cloud: Wormhole and *RG* variants (constrained/unconstrained) across training set sizes of 10, 50, 100 and 200.

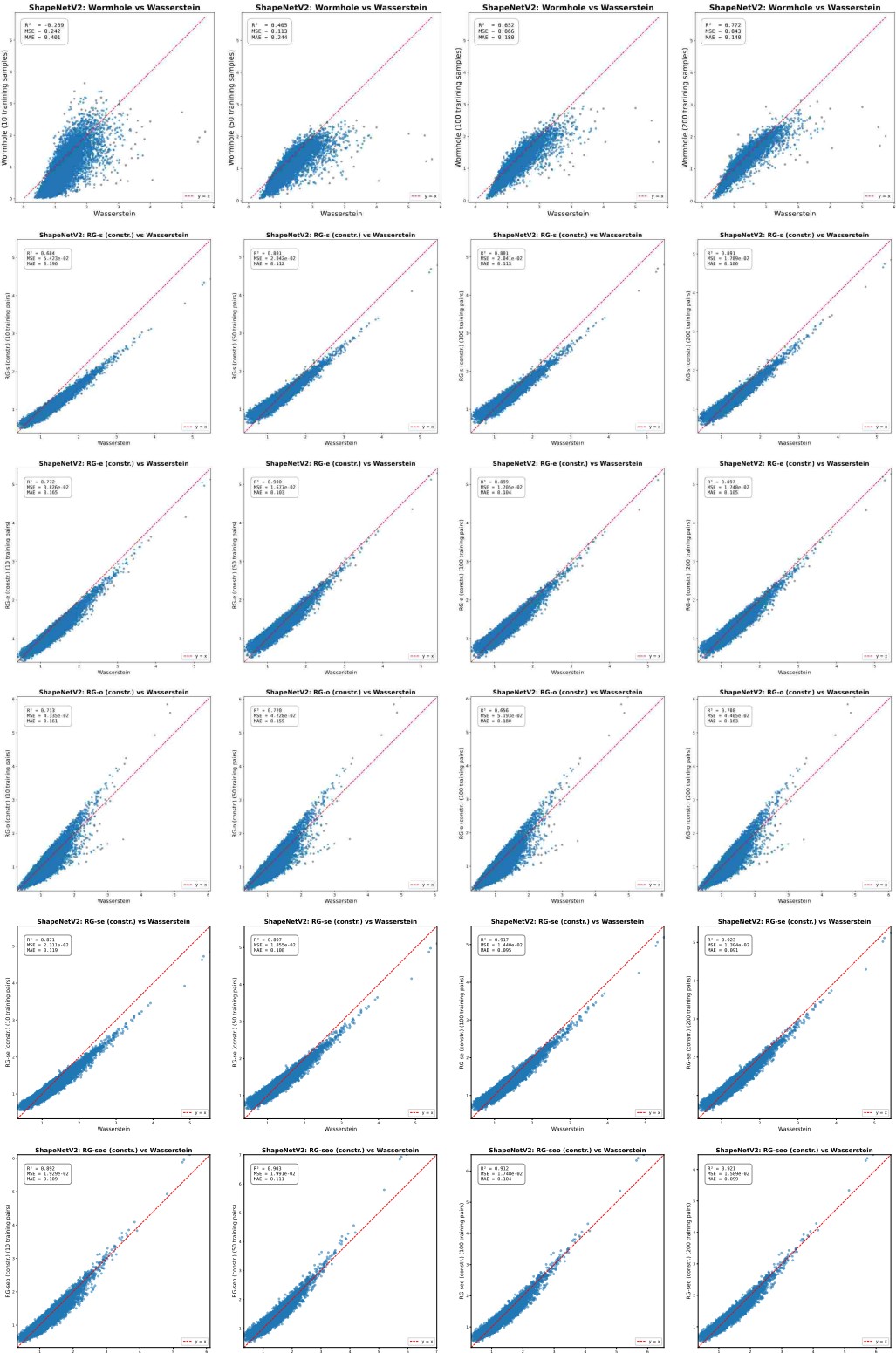

Figure 8: ShapeNetV2 Point Cloud: Wormhole and *RG* variants (constrained/unconstrained) across training set sizes of 10, 50, 100, and 200.

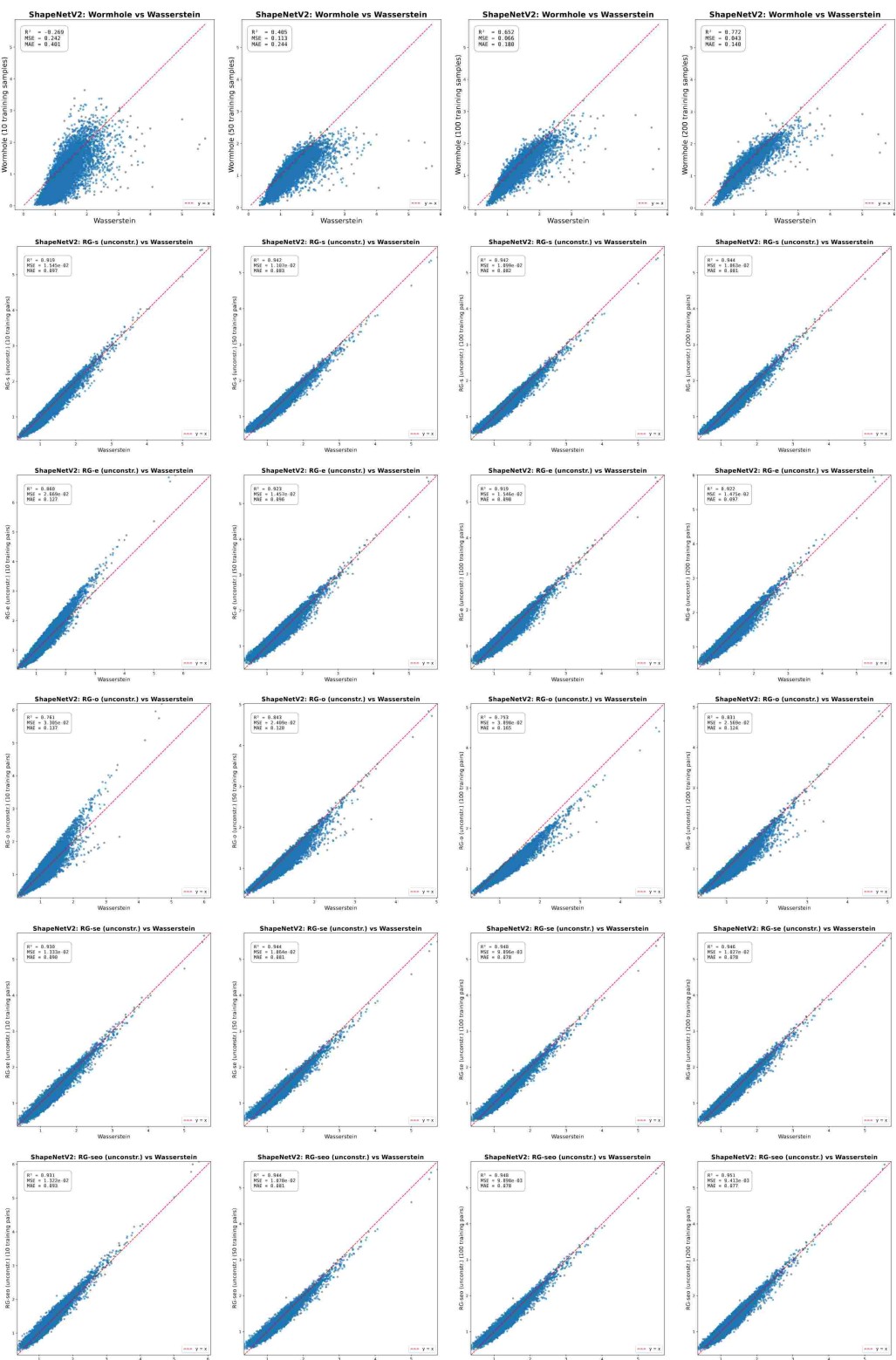

Figure 9: ShapeNetV2 Point Cloud: Wormhole and *RG* variants (constrained/unconstrained) across training set sizes of 10, 50, 100, and 200.

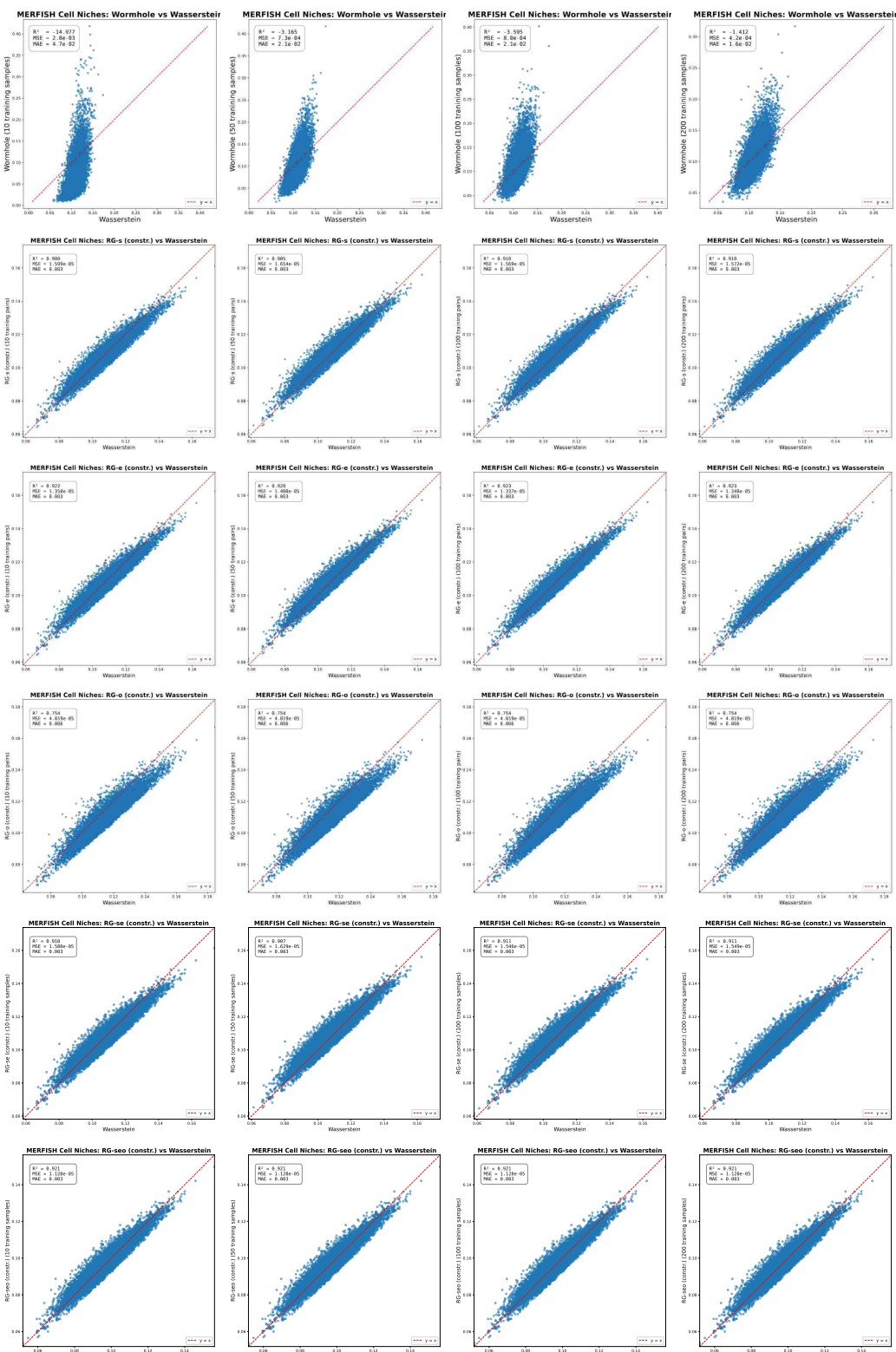

Figure 10: MERFISH Cell Niches: Wormhole and *RG* variants (constrained/unconstrained) across training set sizes of 10, 50, 100, and 200.

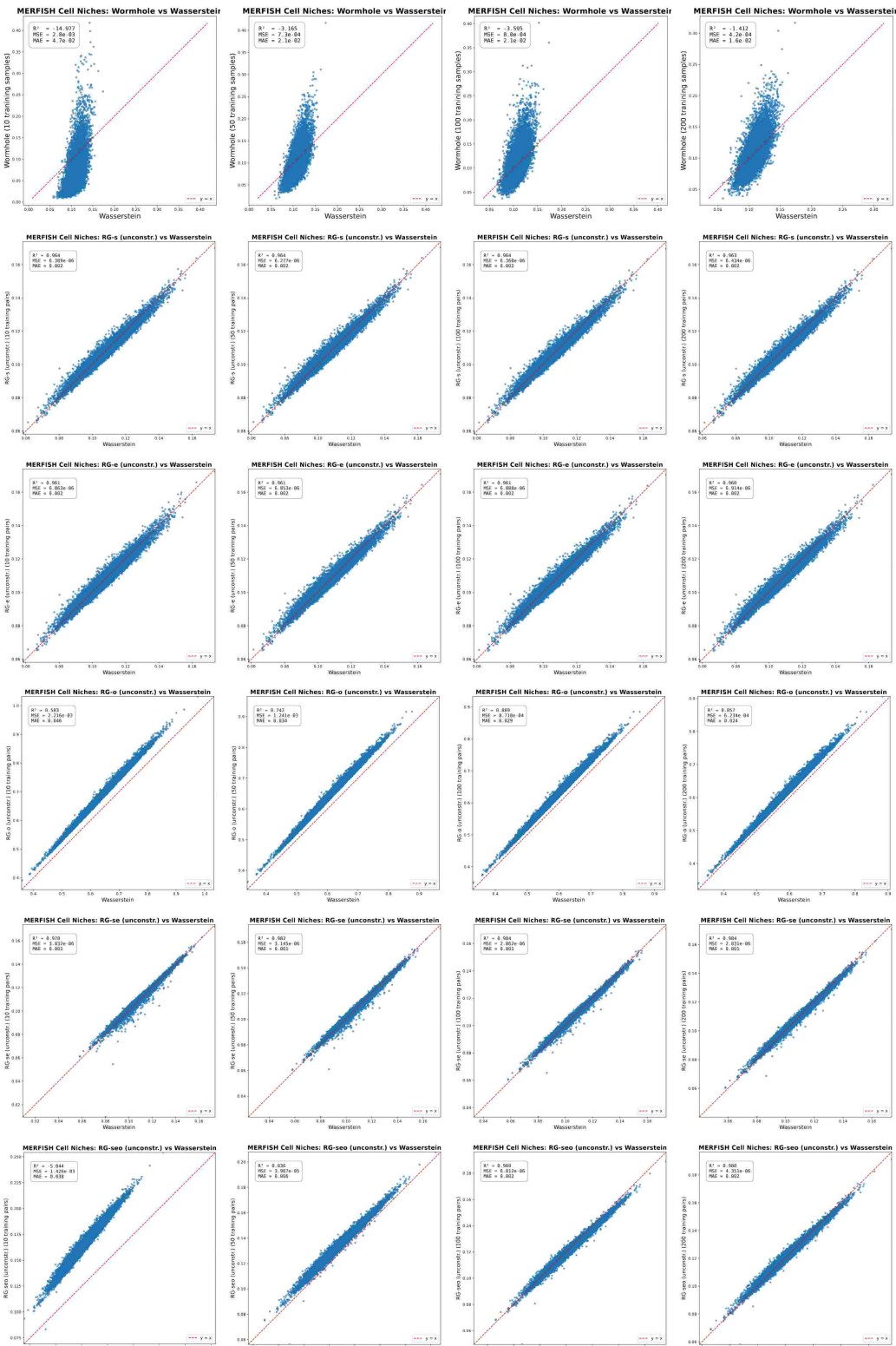

Figure 11: MERFISH Cell Niches: Wormhole and *RG* variants (constrained/unconstrained) across training set sizes of 10, 50, 100, and 200.

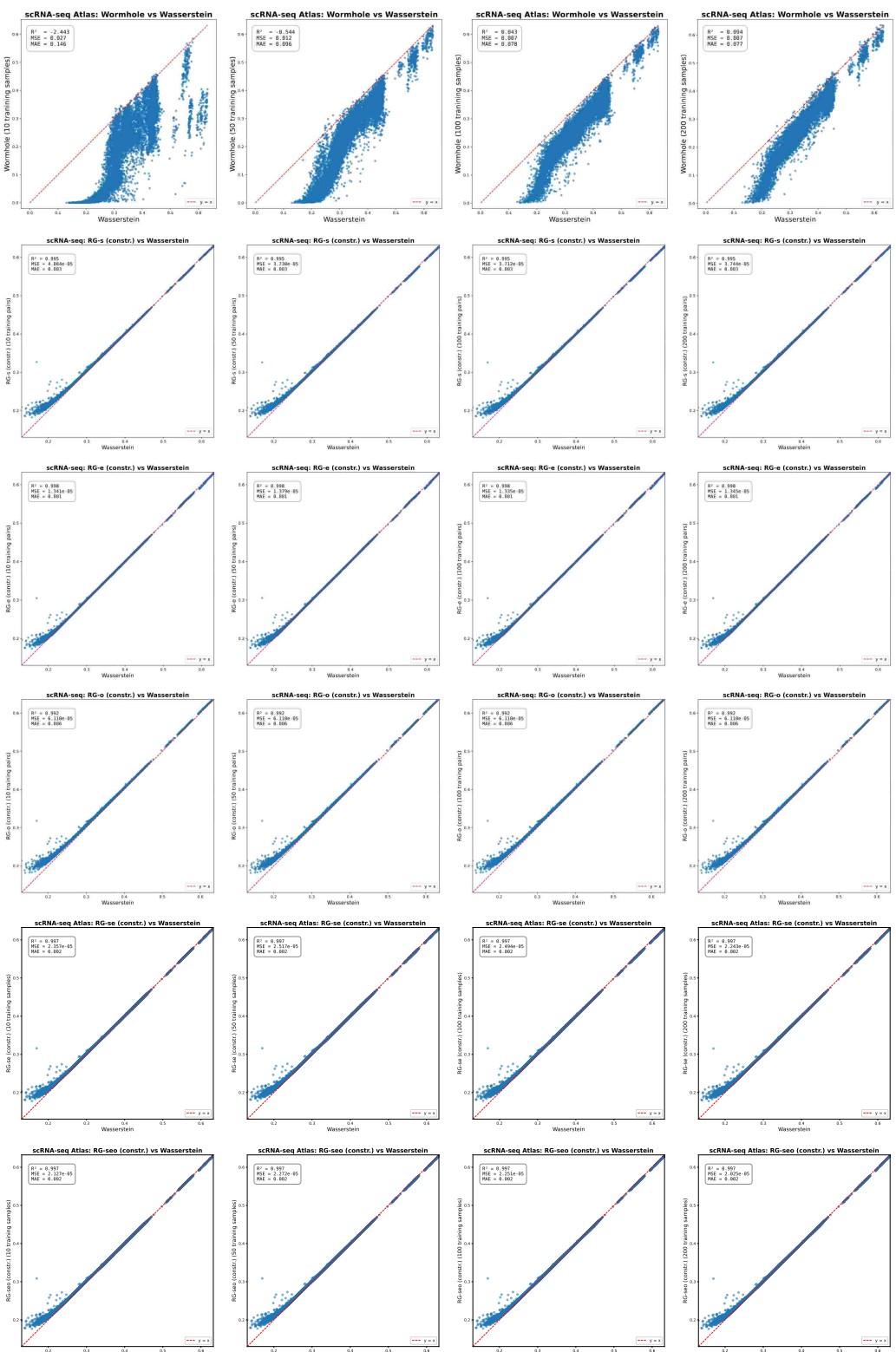

Figure 12: scRNA-seq: Wormhole and *RG* variants (constrained/unconstrained) across training set sizes of 10, 50, 100, and 200.

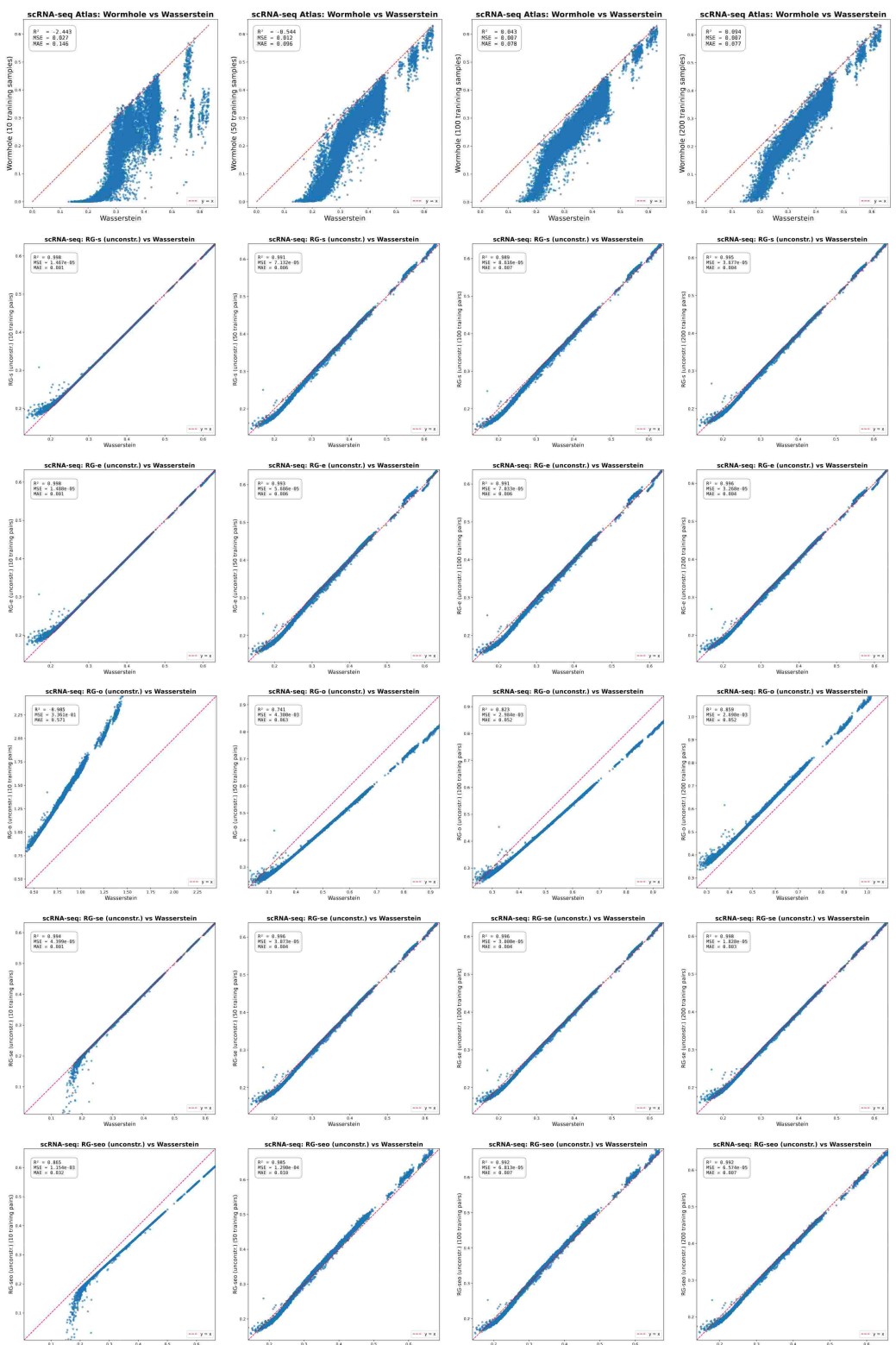

Figure 13: scRNA-seq: Wormhole and *RG* variants (constrained/unconstrained) across training set sizes of 10, 50, 100, and 200.

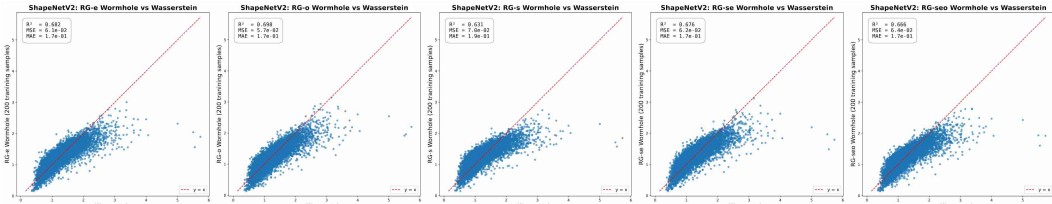

Figure 15: ShapeNetV2: *RG-Wormhole* (constrained model) vs. Wormhole.

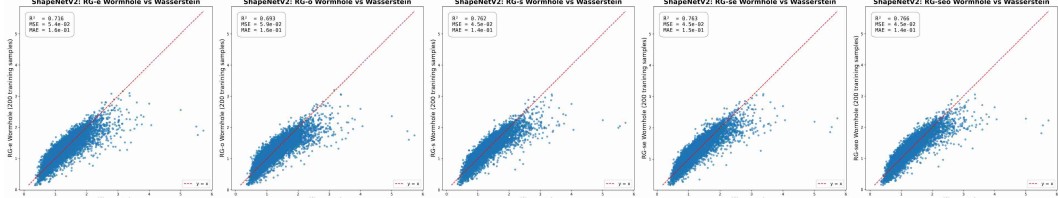

Figure 16: ShapeNetV2: *RG-Wormhole* (unconstrained model) vs. Wormhole

## B.5 RG-Wormhole: Accelerating Wormhole with Regression of Wasserstein

**Experimental Settings.** We run five experiments to show that *RG-Wormhole* is much faster than Wormhole with similar effectiveness. First, we measure training time by training both models under the same optimizer and schedule, sweeping batch sizes from 4 to 20 and reporting wall-clock time for training sets of 10, 50, 100, and 200 pairs. Second, we assess encoders by computing $R^2$/MSE/MAE between pairwise distances in the learned embedding space and exact Wasserstein. Third, we evaluate decoders by reporting the Wasserstein loss between each input and its reconstruction. Fourth, we examine barycenters by decoding the mean embedding of each class and visualizing results. Fifth, we study interpolation by decoding linear paths between two embeddings and illustrating trajectories. Across all experiments, hyperparameters match Wormhole; the only change in *RG-Wormhole* is replacing Wasserstein in encoder and decoder losses with the calibrated unconstrained *RG*. We use 10 samples from the training set to estimate RG coefficients. Except for embedding experiment which uses ShapeNetV2 dataset, other experiments use ModelNet40 dataset, same as (Haviv et al., 2024).

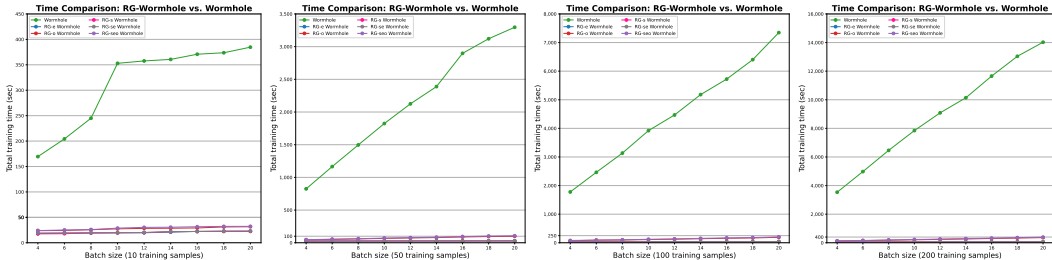

Figure 14: Training time comparison of Wormhole and *RG-Wormhole* methods on point cloud datasets with varying number of training samples.

## B.6 Comparison between RG and classical approximation methods.

**Experimental Settings.** We conducted direct runtime and accuracy comparisons between our methods and two widely used classical OT accelerations, Sinkhorn and Linear OT, under both single-pair computation and large-scale pairwise computations. For Sinkhorn and Linear OT, no training phase is required. For example, in RG-s, we estimate the optimal regression weight using 5 randomly selected samples ($M = 10$ training pairs). The training cost (2.891 seconds (s)) is not included in the inference-time table below, we focus on inference time only. For Sinkhorn, we conduct a grid search over the entropic regularization hyperparameter in {0.01, 0.05, 0.1} and the number of

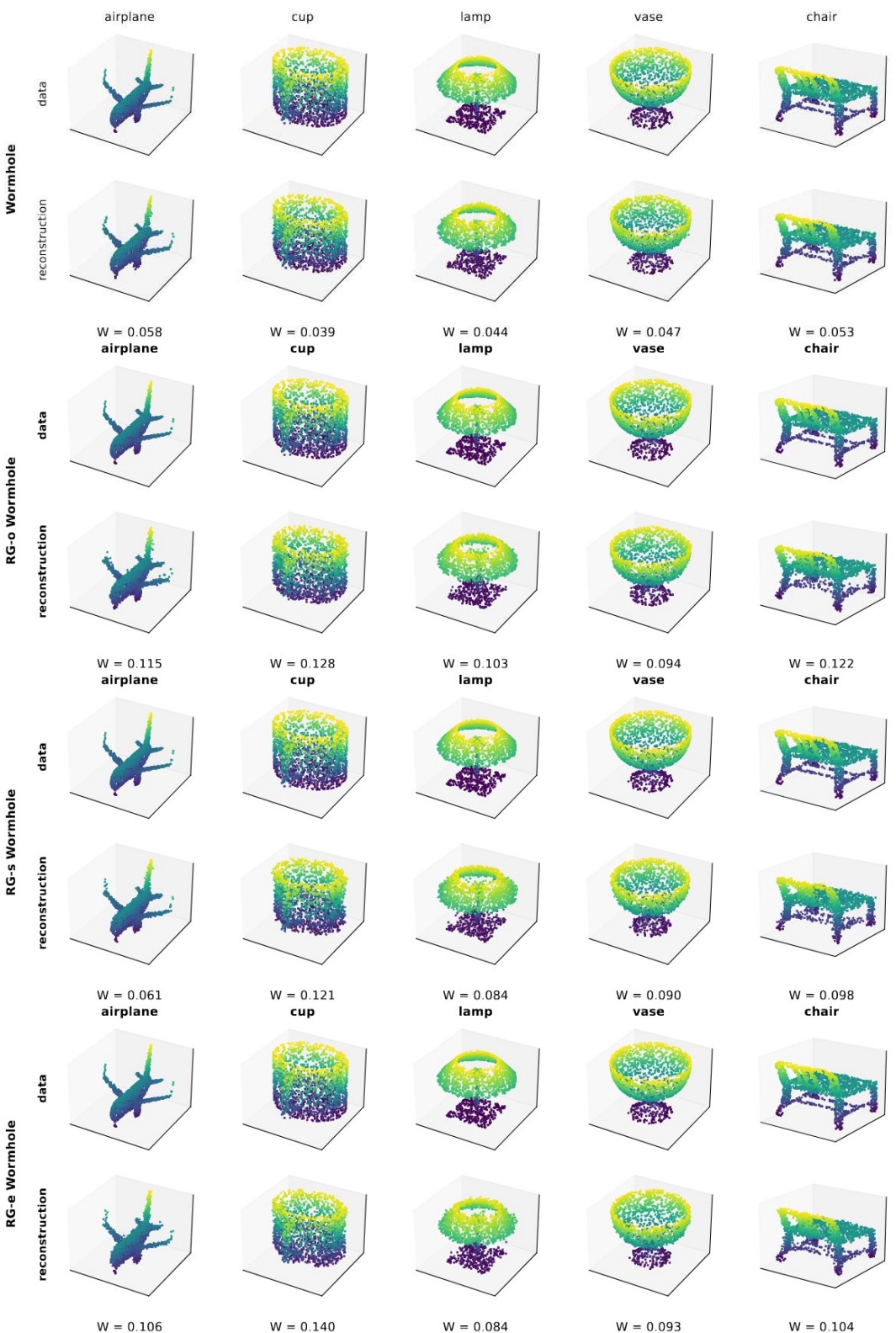

Figure 17: ModelNet40: *RG-Wormhole* vs Wormhole reconstruction experiment.

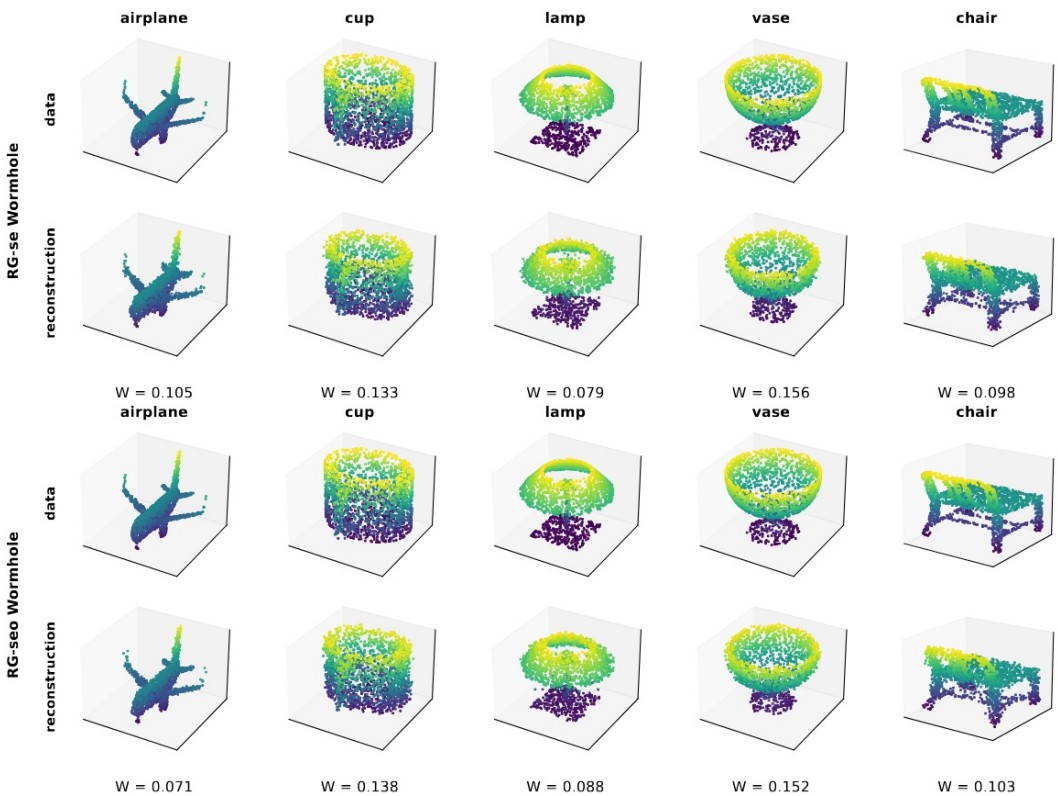

Figure 18: ModelNet40: *RG-Wormhole* reconstruction experiment.

iterations in {500, 1000}, and report the best-performing configuration for each dataset. We do not exceed 1,000 iterations because the runtime already surpasses that of exact Wasserstein.

For **inference runtime comparison**, we use point clouds from ShapeNetV2, where each object consists of 2,048 points in 3D, and measure inference time as the number of pairwise computations in seconds increases (see Table 5). These time measurements are relative and may not be precise due to system conditions.

For **performance comparison**, we evaluate each method's approximation quality relative to exact Wasserstein using $R^2$ in Table 6. Each value is computed over 5,000 test pairs across four datasets: MNIST point clouds, ShapeNetV2, MERFISH cell niches, and scRNA-seq.

**Results.** From the results above, several observations are clear. First, Sinkhorn performs the worst among the compared methods. Even with 1,000 iterations, it fails to converge on MNIST point clouds and ShapeNetV2 (negative $R^2$), while also being slower than both Linear OT and RG-s. However, this may be due to a suboptimal choice of the entropic regularization parameter. In theory, the time complexity of Sinkhorn is $\mathcal{O}((N-M)^2 n^2/\epsilon)$ for $(N-M)^2$ pairs. Therefore, Sinkhorn is computationally expensive in this setting. Second, Linear OT provides reasonable accuracy and moderate speed, but its approximation quality is consistently below that of RG-s, and its runtime grows much faster as the number of distributions increases. As discussed, the time complexity of Linear OT is $\mathcal{O}((N-M)n^2/\epsilon)$. Third, the RG methods provide more accurate estimates than both Sinkhorn and Linear OT. In addition, RG-s, RG-e, RG-o, and RG-seo are all faster than Linear OT. RG-seo achieves a speed comparable to Linear OT while offering better approximation quality, whereas optimization-based SW variants require substantially more computation time.

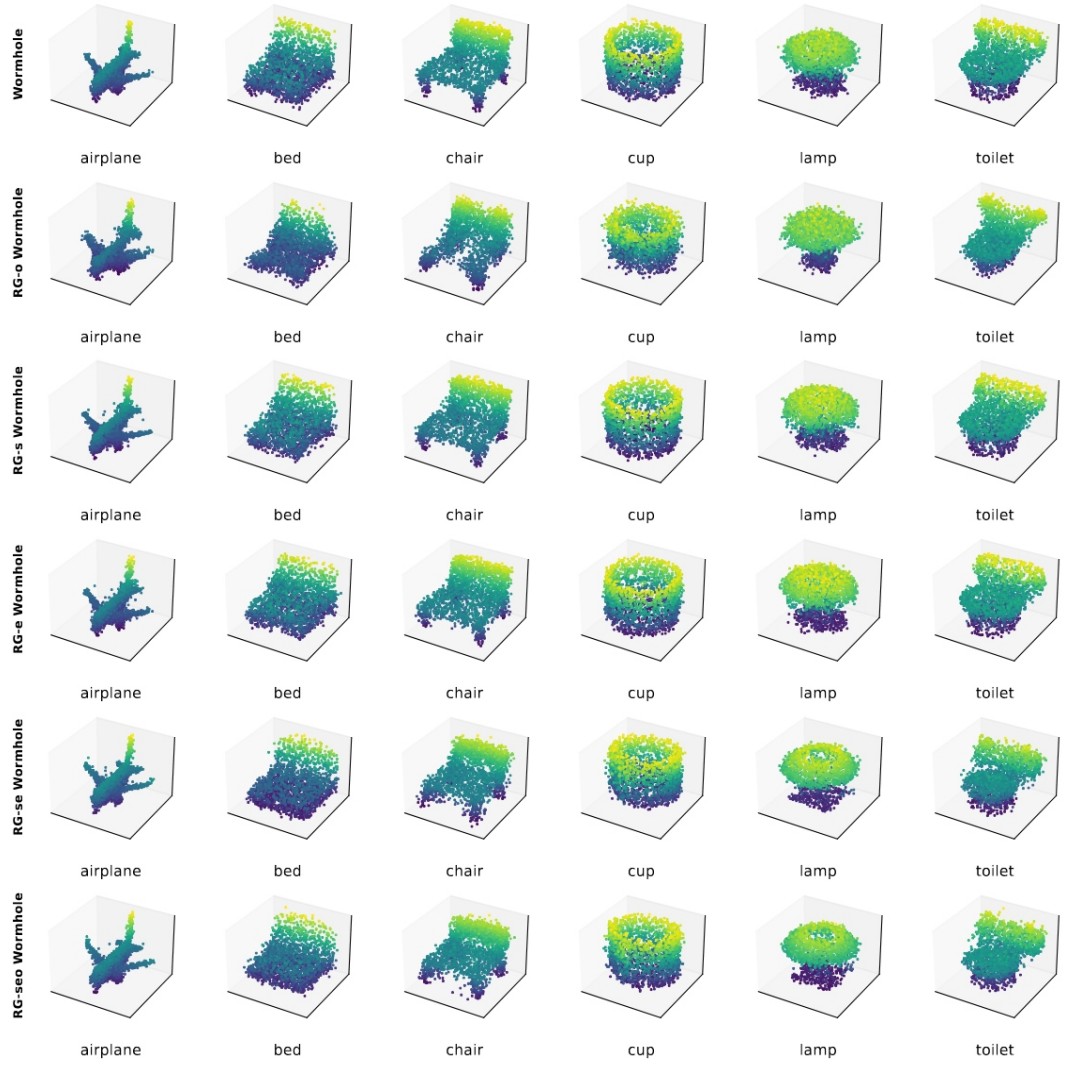

Figure 19: ModelNet40: *RG-Wormhole* barycenter experiment.

Table 5: Comparison in inference runtime among *RG* and classical approximation methods.

| Method | 2 Samples (1 pair) | 10 Samples (45 pairs) | 50 Samples (1,225 pairs) | 100 Samples (4,950 pairs) | 200 Samples (19,900 pairs) |
|---|---|---|---|---|---|
| Wasserstein | 0.604s | 13.351s | 354.823s | 1,427.276s | 5,030.966s |
| Sinkhorn (iters=500) | 0.441s | 10.444s | 286.203s | 1,154.826s | 4,628.393s |
| Sinkhorn (iters=1,000) | 0.482s | 21.012s | 572.441s | 2,308.735s | 9,336.621s |
| Linear OT | 0.634s | 3.141s | 15.473s | 31.121s | 92.134s |
| RG-s | 0.002s | 0.090s | 2.043s | 8.105s | 21.408s |
| RG-e | **0.0005s** | **0.023s** | **1.737s** | **7.060s** | **18.429s** |
| RG-o | 0.013s | 0.136s | 6.634s | 15.310s | 54.900s |
| RG-se | 0.002s | 0.113s | 3.780s | 15.170s | 39.950s |
| RG-seo | 0.013s | 0.147s | 7.017s | 28.840s | 94.200s |

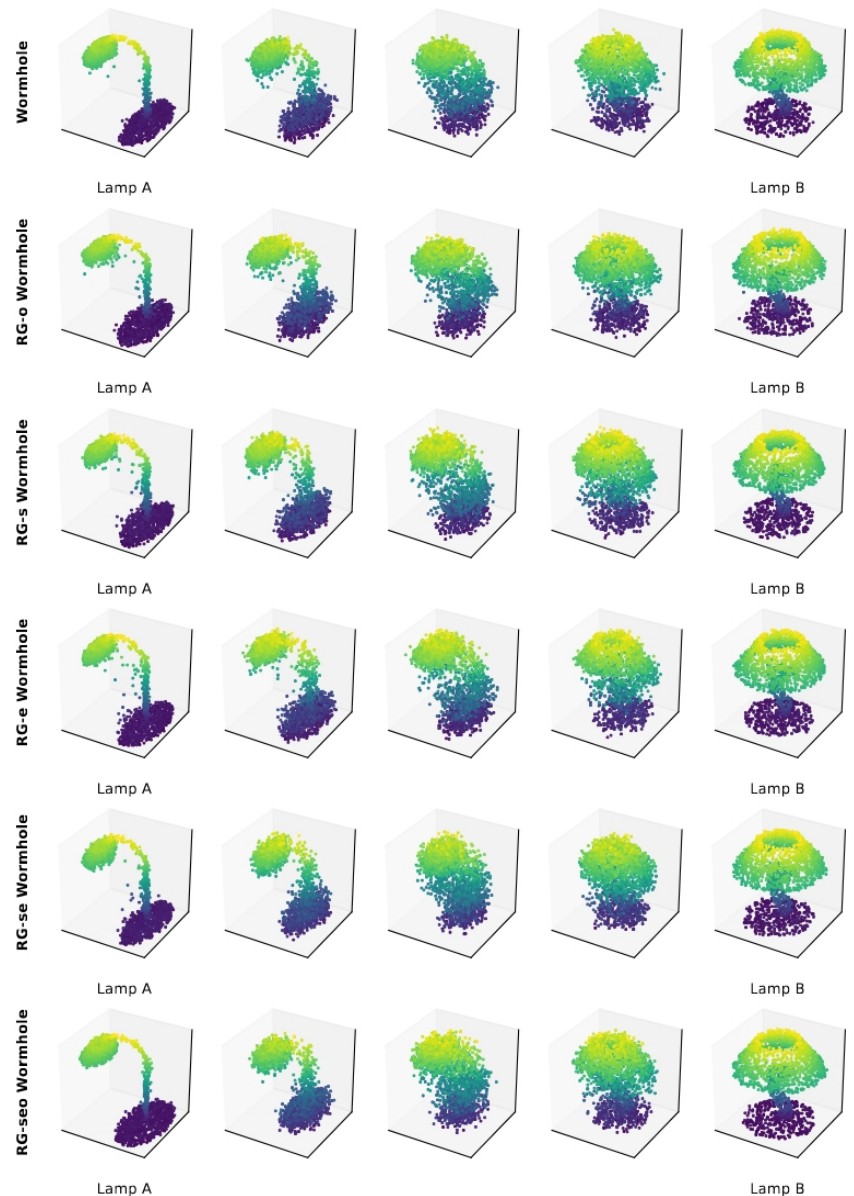

Figure 20: ModelNet40: *RG-Wormhole* barycenter experiment.

Table 6: Comparison in performance among *RG* and classical approximation methods.

| Method | MNIST Point Cloud (10,000 test pairs) | ShapeNetV2 (10,000 test pairs) | MERFISH Cell Niches (10,000 test pairs) | scRNA-seq (10,000 test pairs) |
|---|---|---|---|---|
| Sinkhorn | -1.970 | -1.549 | 0.764 | 0.964 |
| Linear OT | 0.697 | 0.918 | 0.842 | 0.984 |
| RG-s | 0.925 | 0.942 | 0.964 | 0.989 |
| RG-e | 0.910 | 0.919 | 0.961 | 0.991 |
| RG-o | 0.767 | 0.753 | 0.809 | 0.923 |
| RG-se | 0.933 | 0.948 | 0.984 | **0.996** |
| RG-seo | **0.934** | **0.948** | **0.969** | 0.992 |

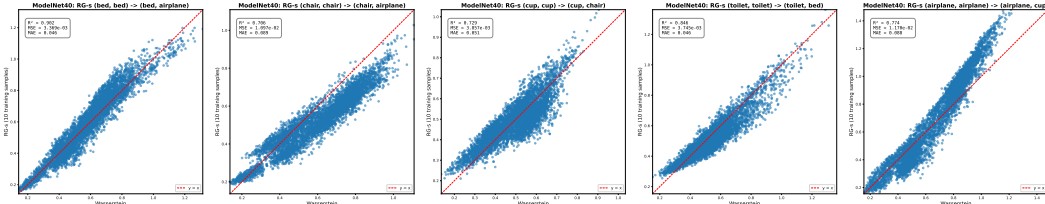

Figure 21: ModelNet40: Intra-class Experiment.

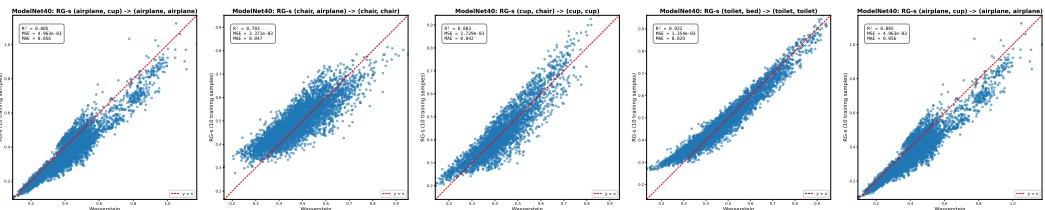

Figure 22: ModelNet40: Inter-class Experiment.

## B.7 EXPERIMENT OF INTER-CLASS AND INTRA-CLASS FOR RG.

**Experimental Settings.** We conducted an experiment to test how well RG generalizes across intra-class and inter-class pairs. We trained RG only on intra-class pairs and tested on inter-class pairs, and vice versa, using five scenarios for each setting on ShapeNetV2.

Concretely, for the *intra-class → inter-class* setting, $(A, A) \to (A, B)$, we ran five experiments. In each experiment, we sampled 10 intra-class training pairs from $(A, A)$, estimated the regression weight, and then constructed a test set by sampling 50 objects from class $A$ and 50 objects from class $B$. This yields 100 distributions in total and therefore 4,950 inter-class test pairs $(A, B)$.

For the *inter-class → intra-class* setting, $(A, B) \to (A, A)$, we again ran five experiments. In each experiment, we sampled five objects from class $A$ and objects from class $B$ to form 10 inter-class training pairs $(A, B)$, estimated the regression weight, and then built a test set by sampling 100 objects from class $A$, which yields 4,950 intra-class test pairs $(A, A)$.

**Results.** We present the results in Figures 21–22. Although this is intentionally suboptimal (the regression weight is learned from a restricted subset of pairs), RG still achieves reasonably high $R^2$ (e.g., 0.902, 0.846, 0.932, 0.805), compared to about 0.95 when training on the full pair distribution. This shows that RG **generalizes well** to out-of-sample data, with performance degradation that is present but moderate.

## B.8 EXPERIMENT OF OPTIMAL WEIGHTS OF RG.

**Results.** We visualized the estimated coefficients of the RG methods for all applications in Figures 23–30. The coefficient patterns vary substantially across datasets, indicating that there is **no fixed set of coefficients** that universally applies to all cases. This further underscores the need for the proposed regression framework, which captures the dataset-specific uncertainty in the association between Wasserstein and sliced Wasserstein distances (i.e., the meta-distribution) on the full pair distribution.

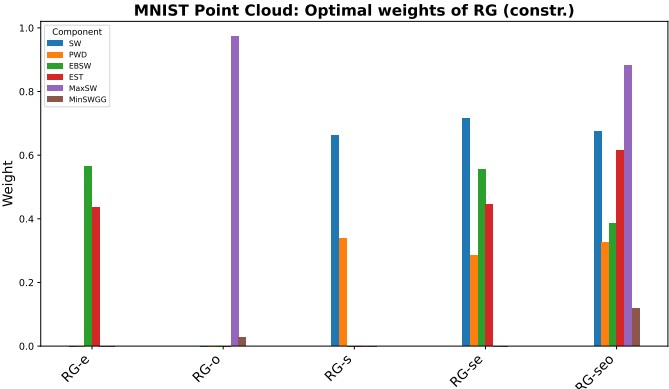

Figure 23: MNIST Point Cloud: Optimal weight of *RG* variants (constrained) across different training samples.

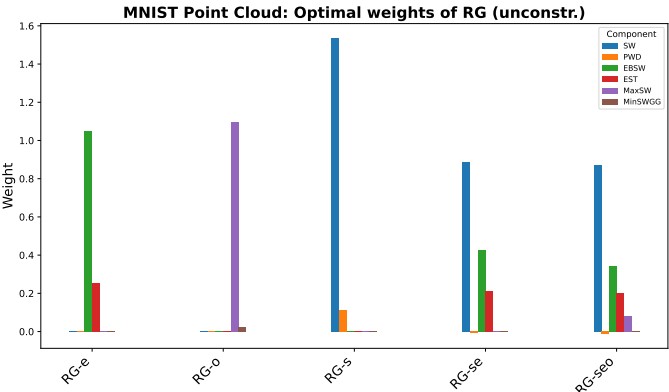

Figure 24: MNIST Point Cloud: Optimal weight of *RG* variants (unconstrained) across different training samples.

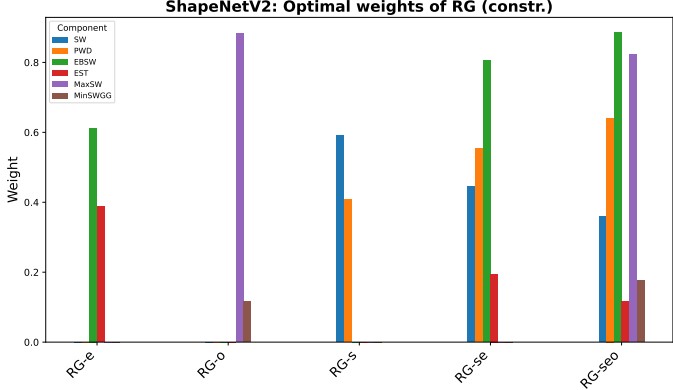

Figure 25: ShapeNetV2: Optimal weight of *RG* variants (constrained) across different training samples.

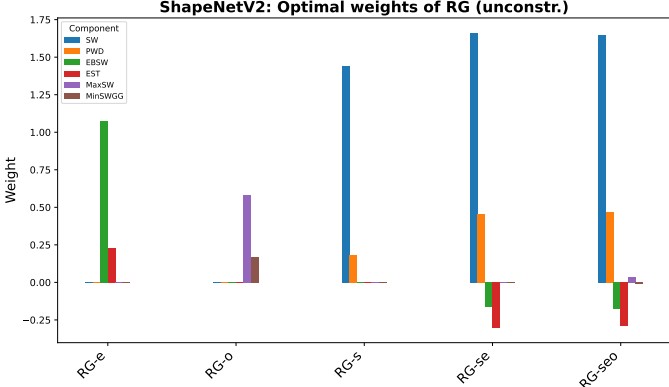

Figure 26: ShapeNetV2: Optimal weight of *RG* variants (unconstrained) across different training samples.

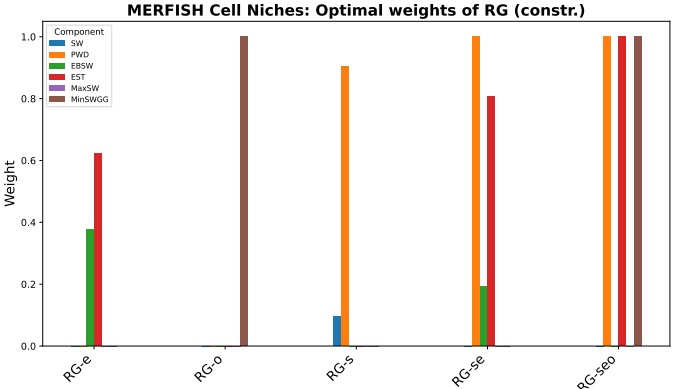

Figure 27: MERFISH Cell Niches: Optimal weight of *RG* variants (constrained) across different training samples.

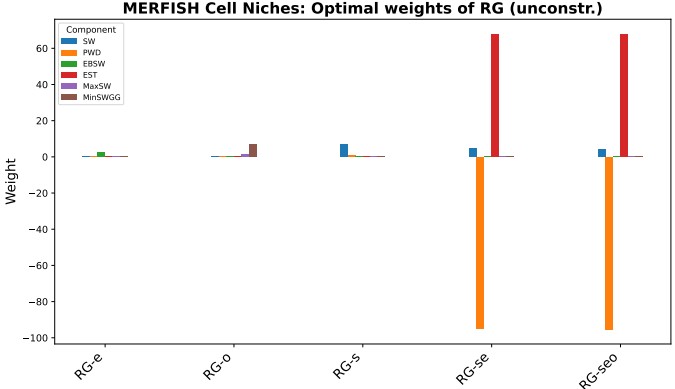

Figure 28: MERFISH Cell Niches: Optimal weight of *RG* variants (unconstrained) across different training samples.

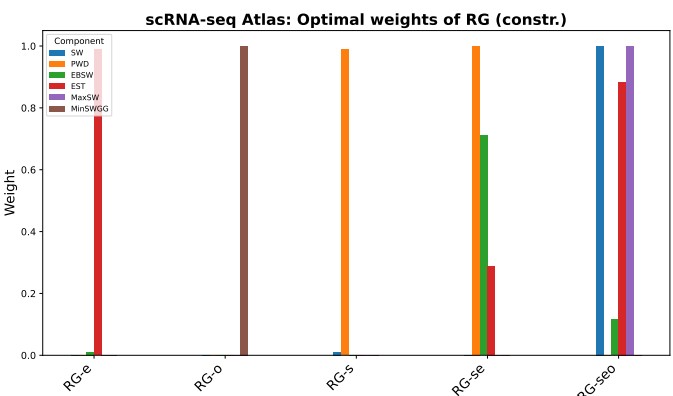

Figure 29: scRNA-seq Atlas: Optimal weight of *RG* variants (constrained) across different training samples.

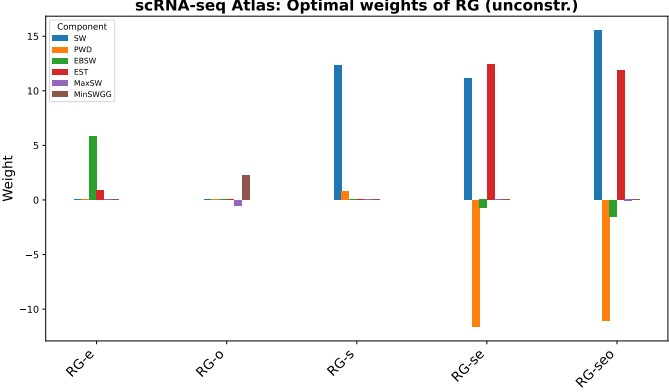

Figure 30: scRNA-seq Atlas: Optimal weight of *RG* variants (unconstrained) across different training samples.

