# OpenReview forum: "Fast Estimation of Wasserstein Distances via Regression on Sliced Wasserstein Distances"
_ICLR.cc/2026/Conference — ICLR 2026 Poster_

### Official Review · Reviewer_A3f7 · 2025-10-29

**Soundness:** 3
**Presentation:** 3
**Contribution:** 2
**Rating:** 2
**Confidence:** 4

**Summary:**

This paper proposed a method to estimate the Wasserstein distance efficiently. Specifically, the proposed method utilizes the fact that the sliced Wasserstein distance is the lower bound of the Wasserstein distance, and the lifted sliced Wasserstein distance is the upper bound of the Wasserstein distance. Then, the proposed method is the weighted average of the sliced Wasserstein distance and the lifted Wasserstein distance and provides a better estimation of the Wasserstein distance by learning these weights.

**Strengths:**

* This paper is well-written and easy to follow.
* The proposed method is simple, and it is intuitively correct that the proposed method is a better estimation of the Wasserstein distance than the sliced Wasserstein distance and the lifted Wasserstein distance.

**Weaknesses:**

* The proposed method is incremental and trivial. It is just a linear combination of the sliced Wasserstein distance and the lifted sliced Wasserstein distance.
* The experimental results are not comprehensive. The main motivation of the proposed method is to efficiently estimate the Wasserstein distance. However, this paper did not evaluate and discuss this efficiency, e.g., the time required to compute, the time required for training.
* There are a lot of papers that proposed the efficient approximation of the Wasserstein distance, but this paper did not compare the proposed method with them in the experiments. Specifically, it is at least necessary to compare the proposed method with the sliced Wasserstein distance, the lifted sliced Wasserstein distance, the max-sliced Wasserstein distance, and other types of approximation of the Wasserstein distance.

**Questions:**

See the weakness section.

---

> ### Author Response · Authors · 2025-11-19
> **Response to Reviewer A3f7 (1)**
>
> We sincerely appreciate the reviewer’s time and thoughtful feedback. We have carefully considered the comments and provide our responses to the questions below.
>
> **Q19.** The proposed method is incremental and trivial. It is just a linear combination of the sliced Wasserstein distance and the lifted sliced Wasserstein distance.
>
> **A19.** Thank you very much for your commemts. We would like to refer the reviewer to the global rebuttal for an explaination about the position of our work in the literature. As discussed in the global rebuttal, our focus is not on introducing linear combination of sliced Wasserstein (SW) variants, but on setting up a regression frameworks of Wasserstein distances onto sliced Wasserstein (SW) variants. As mentioned  Reviewer **txMr**: *“The method is elegant, principled, and provides a practical way to ‘bootstrap’ a collection of cheap estimates to learn a high-quality one.”*).  In particular, we consider the computing Wasserstein distances for **multiple-pairs of distributions** drawn from a meta distribution. We propose a regression framework (Equation (7))  that models the Wasserstein distance as a function of several sliced Wasserstein type distances. This framework is novel as it provides the first principled approach that connects Wasserstein distances to sliced Wasserstein variants. While there are many sliced optimal transport (SOT) metrics, to the best of our knowledge, there is no sliced optimal transport metric that can approximiate Wasserstein distance. Practical use of SOT metric is for an efficient **alterative** of Wasserstein distance.
>
> With the regression framework, we can estimate a model from a subset of pairs of distribution then predict the rest with only the same computation complexies of SW variants.Empirically, our approach is both simple and effective. Across Gaussian mixtures, ShapeNetV2 classification, MERFISH, and scRNA-seq, our RG variants consistently achieve higher approximation quality than Wormhole, a SOTA Wasserstein approximation method, in the low-data regime, while remaining substantially faster than all compared Wasserstein approximations when many distribution pairs are evaluated. Furthermore, when used to accelerate Wormhole (RG-Wormhole), our estimator preserves its embedding quality yet significantly reduces training time. This directly addresses practical bottlenecks in large-scale OT applications, which existing methods struggle with.
>
>
> **Q20.** The experimental results are not comprehensive. The main motivation of the proposed method is to efficiently estimate the Wasserstein distance. However, this paper did not evaluate and discuss this efficiency, e.g., the time required to compute, the time required for training.
>
> **A20.** We appreciate the reviewer’s concern regarding the scope of our efficiency evaluation. We refer the reviewer to **GA2**, where we compare the RG variants with Sinkhorn and Linear OT, demonstrating that our regression framework is substantially faster than both methods while achieving better approximation of Wasserstein distances. We also direct the reviewer to the global rebuttal **GA6**, which provides a detailed training-time comparison between the RG variants and the Wormhole baseline. In summary, RG trains considerably faster than Wormhole while attaining comparable performance in the low-data regime. This improvement arises because RG is a simple linear regression model, whereas Wormhole is a deep Transformer-based architecture that is much more computationally expensive to train.
>
> We further clarify how the proposed approach can accelerate the training of Wasserstein Wormhole. For the hybrid method (RG-Wormhole), we include a detailed training-time comparison in Appendix B.5 (Figure 14). As shown in the figure, RG-Wormhole consistently reduces training time across all evaluated settings while essentially preserving the baseline performance (see Appendix B.5 for details).
>
> We hope these clarifications fully address the reviewer’s concerns.

---

> > ### Author Response · Authors · 2025-11-19
> > **Response to Reviewer A3f7 (2)**
> >
> > **Q21.** There are a lot of papers that proposed the efficient approximation of the Wasserstein distance, but this paper did not compare the proposed method with them in the experiments. Specifically, it is at least necessary to compare the proposed method with the sliced Wasserstein distance, the lifted sliced Wasserstein distance, the max-sliced Wasserstein distance, and other types of approximation of the Wasserstein distance.
> >
> >
> > **A21.** Thank you very much for your comments. As noted in our earlier responses, we are not aware of any sliced optimal transport (SOT) variant that directly approximates the Wasserstein distance. If such a method exists, we would greatly appreciate the reviewer pointing it out. Our focus is on approximating Wasserstein distances in the multiple-pairs setting, rather than the single-pair setting. Any single-pair Wasserstein approximation can, however, be used during the estimation stage of the regression model.
> >
> > We also note that we have already compared our method with multiple SW-based variants. Specifically, the point-cloud classification experiments in Table 3 (Appendix B.2) and the metric-space visualizations in Figure 5 (Appendix B.3) demonstrate that our regression framework provides more accurate estimates of Wasserstein distances than existing SW variants. Consequently, it delivers improved performance on downstream tasks and produces estimates that are closer to the exact Wasserstein distances while retaining the computational efficiency characteristic of SW methods.
> >
> > Furthermore, as discussed in response to a previous question, we conducted additional experiments comparing RG with other classical approximation methods, namely Sinkhorn and Linear OT. We report both inference runtime and approximation quality, and refer the reviewer to the global rebuttal **GA2** for details. These results show that our approach outperforms all baselines, achieving the fastest runtime and the highest accuracy, with particularly significant gains in large-scale pairwise computations. We emphasize again that our method is specifically designed for the multiple-pairs setting (e.g., computing pairwise distances among many distributions), and any single-pair Wasserstein approximation can be seamlessly integrated into our framework to further accelerate regression model estimation.

---

### Official Review · Reviewer_txMr · 2025-10-31

**Soundness:** 4
**Presentation:** 3
**Contribution:** 3
**Rating:** 6
**Confidence:** 5

**Summary:**

In many modern machine learning problems, the datasets are collections of distributions, i.e. a meta-distribution over distribution. In those cases, the Wasserstein distance (WD) is an alluring metric for analysis, but constitutes and significant bottleneck in applications that require many pairwise comparisons. This paper proposes a method to estimate the WD by learning a mapping from computationally cheap Sliced Wasserstein (SW) variants to the true WD. The learning process relies on a small number of "ground-truth" WD calculations to fit the estimator, which can then be applied to new pairs.

The proposed method regresses the true WD onto a set of its sliced variants. The predictors include SW-based distances that are known lower bounds (e.g., standard SW, Max-SW) and "lifted" SW distances that serve as upper bounds (e.g., Projected Wasserstein, Min-SWGG). The paper details two linear models for this regression: an unconstrained model with a closed-form least-squares solution, and a constrained model. The constrained model pairs lower and upper bounds, reducing the number of parameters and adding an inductive bias, which is intended for "few-shot" learning scenarios with limited ground-truth samples.

This regression estimator (dubbed "RG") is evaluated in two applications. First, as a standalone estimator, it is applied to a k-NN classification task, where its accuracy is reported to be close to that of the exact WD. The paper also shows it achieves higher accuracy than the Wasserstein Wormhole model in low-data regimes. Second, the authors propose "RG-Wormhole," a hybrid that replaces the expensive WD calculations in the Wormhole training loop with the fast RG estimate. This substitution is shown to reduce training time substantially while reportedly maintaining comparable performance on tasks like point cloud reconstruction and interpolation

**Strengths:**

The method is elegant, principled, and provides a practical way to "bootstrap" a collection of cheap estimates to learn a high-quality one. The approach is clearly effective, as demonstrated by extensive comparisons and benchmarks across multiple datasets and tasks (k-NN, embedding, reconstruction). Furthermore, the data efficiency is a major plus as the ability to learn a good regression model from very few "ground-truth" Wasserstein distance calculations makes this highly applicable. Finally, the application within the "RG-Wormhole" model is a strong contribution, as it directly addresses the primary computational bottleneck of a current method.

**Weaknesses:**

The presentation of "Propositions" 1 and 2 (Section 3.2, Page 6) is a notable weakness. These appear to be standard, well-known closed-form solutions for unconstrained and constrained linear regression, respectively. Presenting these basic results as "propositions" with proofs in the appendix feels unnecessary and could be interpreted as an attempt to add theoretical weight where none is needed. An algorithmic and empirical paper like this does not require such justifications for using standard linear regression.

**Questions:**

The paper highlights the "explainable" nature of linear regression as a reason for its use. Given this, have the authors analyzed the learned coefficients ($\omega$)? Is there a discernible pattern in the learned weights, such as a consistent difference in those assigned to lower bounds versus upper bounds? Similarly, in the constrained model, is there a consistent pattern in the weights for paired "lifted" (upper) and "unlifted" (lower) SW distances across different datasets or dimensions (e.g., does the model learn to trust one more than the other)? The Gaussian simulation in the appendix (Fig 4) hints at this, but a broader analysis on the real datasets would be valuable.

---

> ### Author Response · Authors · 2025-11-19
> **Response to Reviewer txMr**
>
> We appreciate the reviewer’s time and constructive feedback. Their comments have provided valuable insights, and we would like to address the questions raised as follows.
>
>
> **Q17.**
> The presentation of "Propositions" 1 and 2 (Section 3.2, Page 6) is a notable weakness. These appear to be standard, well-known closed-form solutions for unconstrained and constrained linear regression, respectively. Presenting these basic results as "propositions" with proofs in the appendix feels unnecessary and could be interpreted as an attempt to add theoretical weight where none is needed. An algorithmic and empirical paper like this does not require such justifications for using standard linear regression.
>
>
> **A17.** Thank you very much for your comment. We agree that the closed-forms are well-known solutions in linear regression literature. Our original aims is just provides the proofs for completeness. We will change the propositions into remarks for avoid misinterpretation.
>
>
> **Q18.**
> The paper highlights the "explainable" nature of linear regression as a reason for its use. Given this, have the authors analyzed the learned coefficients ($\omega$)?
> Is there a discernible pattern in the learned weights, such as a consistent difference in those assigned to lower bounds versus upper bounds?  Similarly, in the constrained model, is there a consistent pattern in the weights for paired "lifted" (upper) and "unlifted" (lower) SW distances across different datasets or dimensions (e.g., does the model learn to trust one more than the other)? The Gaussian simulation in the appendix (Fig 4) hints at this, but a broader analysis on the real datasets would be valuable.
>
> **A18.** From Figure 4, the weight for  "unlifted" (lower) SW distances are higher in low-dimension while the weight for "lifted" (upper) SW distances is higher for high-dimension. This is consistent with theoretical results from previous studies about the relationship  between  "lifted" (upper) SW distances [9] with Wasserstein distance i.e., min-SWGG and WD coincide when $d > 2n$.
>
> We added visuallization of the estimated coefficients of the RG methods for all applications in **Figures 21–28 in the Appendix**. The coefficient patterns vary substantially across datasets, indicating that there is **no fixed set of coefficients** that universally applies to all cases. This means that there is no universal set of coefficients for all datasets which further underscores the need for the proposed regression framework, which captures the dataset-specific uncertainty in the association between Wasserstein and sliced Wasserstein distances (i.e., the meta-distribution). We see that the weights of constrained model are more intepretable since they are in $[0,1]$
>
>
> [9] Mahey et al, Fast Optimal Transport through Sliced Wasserstein Generalized Geodesics

---

> > ### Comment · Reviewer_txMr · 2025-11-25
> > **Response to Rebuttal**
> >
> > I thank the authors for their response, it cleared some things up for me. I will keep my (positive) score of this work.

---

> > > ### Author Response · Authors · 2025-11-25
> > > **Official Comments by Authors**
> > >
> > > Dear Reviewer txMr,
> > >
> > > Thank you very much for keeping the positive score of 6 for our paper. We genuinely appreciate your constructive comments. We will incorporate all of the discussed points into the revised version of the paper. Please feel free to let us know if you have any further questions.
> > >
> > > Best regards,
> > >
> > > Authors

---

### Official Review · Reviewer_AafT · 2025-10-31

**Soundness:** 3
**Presentation:** 4
**Contribution:** 3
**Rating:** 6
**Confidence:** 4

**Summary:**

The paper's central hypothesis is that for a given meta-distribution $\mathbb{P}(\mu, \nu)$, the true $W_p$ can be accurately and efficiently modeled as a simple linear function of various sliced Wasserstein distances. The core technical insight is to use as predictors a vector of SW based metrics that includes both established lower bounds (e.g., SW, Max-SW, EBSW) and upper bounds (e.g., PW, Min-SWGG, EST).

The authors claim the weights $\omega$ for this regression $W_p \approx \omega^\top S_p(\mu, \nu)$ are a stable characteristic of $\mathbb{P}$ and can be learned in a few-shot setting by computing a small number of ground-truth $W_p$ pairs. This yields the RG surrogate which is claimed to be far more accurate than any single SW baseline and outperform deep learning estimators (like Wasserstein Wormhole) in low-data regimes.

**Strengths:**

- This is a nicely presented paper. The authors' central (and successful) bet is that for a given problem domain $\mathbb{P}$, $W_p$ doesn't just lie somewhere in this interval, but at a consistent relative position that a simple linear model can capture. Thus moves beyond using SW as a fast proxy and instead models the approximation error by bracketing the true value.

- The paper's primary claim of data-efficiency is validated with solid empirical evidence. Table 2 for example demonstrates that with only 100 training pairs, the SOTA deep baseline (Wormhole) is completely unusable (e.g., $R^2$ of 0.65 on ShapeNetV2, -3.6 on MERFISH, 0.04 on scRNA-seq). In stark contrast, the proposed unconstrained RG variants achieve $R^2$ scores of 0.93-0.99 across all datasets, even in the 2,500-dimensional scRNA-seq domain. This high correlation translates directly to downstream utility. For example in the k-NN classification task (Table 1), the RG-seo surrogate (83.5\% accuracy @ k=5) performs almost identically to the true, expensive $W_p$ (84.2\% accuracy @ k=5), while the best single SW baseline is not competitive.

- By replacing $W_p$ inside the Wormhole training loop the authors achieve training speedup. Figure 14 shows almost flat scaling of compute with batch size for RG-Wormhole compared to Wormhole’s quadratic growth, showing that the surrogate can be dropped into OT-based architectures without much performance loss.

**Weaknesses:**

- The entire method relies on a bootstrap phase where $M$ ground-truth $W_p$ pairs are computed to fit the regression weights $\omega$. The paper frames this as a minor setup cost because $M_0=10$ (yielding $M = \frac{10 \times 9}{2} = 45$ pairs) works for their experiments. However, the paper seems to provide no analysis on how to determine the sufficient $M$ for a new problem.
- OLS is only optimal if the error term $\epsilon$ has constant variance. However figures 6-13 suggests otherwise. Would a simple weighted least-squares (WLS) model yield more accurate surrogate?
- Def 3 adds inductive bias and has half of the parameters, which is often helpful when having limited observed samples. In Table 2 (the $M_0=100$ low-data setting), the unconstrained model is superior in nearly every case (e.g., RG-seo on ShapeNetV2: 0.95 $R^2$ unconstrained vs. 0.92 constrained). This holds even for data size ($M_0=10$) in the appendix (e.g., Fig 8, ShapeNetV2: RG-seo 0.93 unconstrained vs. 0.90 constrained). The empirical justification for the constrained model seems week here.

**Questions:**

- The RG-seo model requires computing 6 SW-based predictors, some of which (Max-SW, Min-SWGG) require their own iterative optimization. The paper's time complexity analysis and RG-Wormhole training plot (Fig 14) do not provide a direct inference-time comparison. What is the wall-clock time (for a single pair) comparing RG-seo (at inference) to a standard, converged entropic-regularized $W_p$ (Sinkhorn) approximation?
- How sensitive is the learned regressor to the choice and number of slicing directions (L) and to the specific mix of lower vs upper SW variants?
- What if we train an RG model only on intra-class pairs (e.g., (chair, chair)) and test its $R^2$ on inter-class pairs (e.g., (chair, airplane))? vice versa?

---

> ### Author Response · Authors · 2025-11-19
> **Response to Reviewer AafT (1)**
>
> We appreciate the reviewer’s time and constructive feedback and would like to address the questions as follows
>
> **Q11.** The entire method relies on a bootstrap phase where $M$ ground-truth $W_p$ pairs are computed to fit the regression weights $\omega$. The paper frames this as a minor setup cost because $M_0 = 10$ (yielding $M = \frac{10 \times 9}{2} = 45$ pairs) works for their experiments. However, the paper seems to provide no analysis on how to determine the sufficient $M$ for a new problem.
>
> **A11.** Thank you very much for your comments. It is well-known that for linear model, we have $\mathbb{E}[\|\omega-\hat{\omega}\|_2^2] = \sigma^2 Trace((\hat{S}^\top \hat{S})^{-1})$  which reduces approximately with the rate of $1/M$ [5] (the exact finte rate depends on the construction of $\epsilon$).  In the paper, we empirically investigate how predictive performance changes as we vary $M_0 \in \{10, 50, 100, 200\}$ across all applications; see **Figures 6–13 in Appendix B.4**. Overall, increasing $M_0$ systematically improves performance, as expected from the above rates. Importantly, on all four datasets, most RG variants already achieve strong performance with $M_0 = 10$: the $R^2$ is high and then improves steadily but moderately as $M_0$ grows, indicating that **a very small bootstrap set is often sufficient for practical use**. The only notable exception is RG-seo, which, especially on high-dimensional datasets such as scRNA-seq, requires more training samples for its weights to converge, but in return achieves the most superior performance among all compared variants.
>
>
>
> Overall, we see the problem of selecting $M_0$ (or $M$) as a budget constraint problem. In particular, we need to compute Wasserstein distance for all $M$ pairs of distributions which are computationally expensive. Therefore, we want to use the largest $M_0$ or $M$ which we can afford computation.  In the paper, we assume that there is no distribution shift. In particular, we only estimate regression model from bootstrap pairs of distributions from a dataset and predict for other pairs in the same dataset. We do not aim to use such estimation for out-of-distributions pairs e.g., from point-clouds to single-cell. We believe the main reason for the proposed regression approach work is that there exists an underlying relationship between Wasserstein distances and (lifted) sliced Wasserstein distances for pairs of distributions that are drawn from a meta-distribution.
>
> [5]  Greene "Econometric Analysis" Chapter 4.
>
>
> **Q12.** OLS is only optimal if the error term $\epsilon$ has constant variance. However, Figures 6–13 suggest otherwise. Would a simple weighted least-squares (WLS) model yield a more accurate surrogate?
>
>
> **A12.** From Figures 6–13, we believe that $\epsilon$ has relative constant variance. Therefore, a linear model works quite well. Again, we believe that the choice of regression model is an opening question. We choose the linear model due to its simplcity. A weighted linear can also be applied, however, choosing the weights lead to a different problems of weighting pairs of distributions.
>
> **Q13.** Def 3 adds inductive bias and has half of the parameters, which is often helpful when having limited observed samples. In Table 2 (the $M_0 = 100$ low-data setting), the unconstrained model is superior in nearly every case (e.g., RG-seo on ShapeNetV2: 0.95 $R^2$ unconstrained vs. 0.92 constrained). This holds even for data size ($M_0 = 10$) in the appendix (e.g., Fig 8, ShapeNetV2: RG-seo 0.93 unconstrained vs. 0.90 constrained). The empirical justification for the constrained model seems weak here.
>
>
> **A13.** Thank you for your insightful comment. The main benefit of the constained model is having a half of the parameters. Moreover, it is more interpreatable since it relies on the well-known propertes of SW variants (lower bounds and upperbounds of Wasserstein distance). We agree that the unconstrained model is better in practice (We have rerun the constained model to verify again). However, we still want to include the constrained model for a complete discussion and potential use in future works.

---

> > ### Author Response · Authors · 2025-11-19
> > **Response to Reviewer AafT (2)**
> >
> > **Q14.** The RG-seo model requires computing 6 SW-based predictors, some of which (Max-SW, Min-SWGG) require their own iterative optimization. The paper’s time complexity analysis and RG-Wormhole training plot (Fig 14) do not provide a direct inference-time comparison. What is the wall-clock time (for a single pair) comparing RG-seo (at inference) to a standard, converged entropic-regularized $W_p$ (Sinkhorn) approximation?
> >
> > **A14.** We refer the reviewer to **GA2** for a detail comparison with Sinkhorn approximation. We see that all RG methods are much faster than Sinkhorn approximation (even better approximation). RG-s, RG-e, RG-se are faster than RG-seo which requires iterative optimization as mentioned by the reviewer. Again, we would like to recall that our goal is approximate Wasserstein distances in multiple-pairs setting. We can also use Sinkhorn for approximate the Wasserstein distances when we estimate the regression model.
> >
> > **Q15.** How sensitive is the learned regressor to the choice and number of slicing directions ($L$) and to the specific mix of lower vs. upper SW variants?
> >
> > **A15.** We refer the reviewer to **GA4** for a detailed sensitivity analysis of the number of slicing directions. In summary, as we increase $L$, the optimal weights converge and the performance becomes better and more stable. Overall, we assume that $L$ is big enough to approximate well SW variants. It is possible to integrate recent works on improving approximation of SW variants [6,7,8].  However, we view our proposed regression method and variance reduction for SW as largely orthogonal directions, and we leave their integration for future work.
> >
> > [6] Nguyen et al, "Sliced Wasserstein Estimation with Control Variates"
> >
> > [7] Leluc et al, "Sliced-Wasserstein Estimation with Spherical Harmonics as Control Variates"
> >
> > [8] Sisouk  et al, "A User's Guide to Sampling Strategies for Sliced Optimal Transport"
> >
> >
> >
> > **Q16.** What if we train an RG model only on intra-class pairs (e.g., (chair, chair)) and test its $R^2$ on inter-class pairs (e.g., (chair, airplane))? Vice versa?
> >
> >
> > **A16.** We thank the reviewer for this interesting question and have performed the requested analysis.
> >
> > For the *intra-class $\rightarrow$ inter-class* setting: ((A, A) $\rightarrow$ (A, B)), we ran 5 experiments. In each experiment, we sampled 10 training pairs from (A, A), estimated the optimal regression weight, and then constructed a test set by sampling 50 objects from class (A) and 50 objects from class (B). This yields 100 distributions in total and therefore 4,950 inter-class test pairs (A, B).
> >
> > For the *inter-class $\rightarrow$ intra-class* setting: ((A, B) $\rightarrow$ (A, A)), we again ran 5 experiments. In each experiment, we sampled 5 objects from (A) and 5 objects from (B) to form 10 inter-class training pairs (A, B), and then built a test set by sampling 100 objects from (A), which yields 4,950 intra-class test pairs (A, A).
> >
> > In both settings, the regression weight is learned on a restricted subset of pairs (only intra-class or only inter-class), rather than on the full “true’’ training distribution. As expected, this leads to a drop in performance compared to training on the full dataset, but the degradation is moderate, indicating reasonable robustness.
> >
> > Concretely, training on the full dataset pattern yields an $R^2$ of about **0.95**. When we train only on intra- or inter-class pairs and test on the complementary pattern, the performance decreases but remains relatively high. For example, $(\text{bed}, \text{bed}) \rightarrow (\text{bed}, \text{airplane})$ achieves $R^2$ = 0.902; $(\text{toilet}, \text{toilet}) \rightarrow (\text{toilet}, \text{bed})$ achieves $R^2$ = 0.846; $(\text{toilet}, \text{bed}) \rightarrow (\text{toilet}, \text{toilet})$ achieves $R^2$ = 0.932; $(\text{airplane}, \text{cup}) \rightarrow (\text{airplane}, \text{airplane})$ achieves $R^2$ = 0.805.
> >
> > These results confirm that, although training on a partial pair distribution (intra- or inter-class only) is suboptimal, the **RG model still generalizes reasonably well** across pair types, with performance gaps that are present but not large. The full results are reported in **Figures 29–30 in the Appendix**.

---

> > > ### Comment · Reviewer_AafT · 2025-11-19
> > >
> > > I truly thank the authors for the detailed response. I have adjusted my score accordingly.
> > >
> > > Best regards,

---

> > > > ### Author Response · Authors · 2025-11-19
> > > >
> > > > Thank you very much for increasing the score We genuinely appreciate your constructive comments. We will incorporate all of the discussed points into the revised version of the paper. Please feel free to let us know if you have any further questions.
> > > >
> > > > Best regards,
> > > >
> > > > Authors

---

### Official Review · Reviewer_mmm4 · 2025-10-31

**Soundness:** 3
**Presentation:** 3
**Contribution:** 2
**Rating:** 4
**Confidence:** 5

**Summary:**

This paper introduces a novel regression-based framework for fast estimation of Wasserstein distances between probability distributions. Instead of directly solving the optimal transport (OT) problem or training deep neural networks (e.g., Wasserstein Wormhole, DWE), the authors propose to regress the true Wasserstein distance on a collection of Sliced Wasserstein (SW) and lifted SW distances, which act as lower and upper bounds respectively.

Two linear regression models are introduced:
Unconstrained model, admitting a closed-form least-squares solution;
Constrained model, which enforces the upper/lower bound structure with half as many parameters.

Once trained on a small set of distribution pairs, the model can predict Wasserstein distances for new pairs at the cost of computing SW distances, significantly reducing computational overhead.

The authors validate the approach on Gaussian mixtures, ShapeNetV2 point clouds, MNIST point clouds, MERFISH cell niches, and scRNA-seq datasets, showing that:

The regression estimator achieves high accuracy even with a few training samples.

It outperforms Wasserstein Wormhole in low-data regimes. Substituting it into Wormhole yields RG-Wormhole, which preserves accuracy while substantially reducing training time.

**Strengths:**

The paper presents a conceptually simple yet impactful framework for approximating Wasserstein distances through regression on sliced Wasserstein (SW) and lifted SW distances.
This formulation is the first to explicitly treat Wasserstein estimation as a supervised learning problem over a meta-distribution of random distribution pairs, bridging the gap between classical OT theory and data-driven surrogate modeling.

**Weaknesses:**

### **Weaknesses**

**1. Limited contribution and unclear computational advantage**
The main technical novelty of this paper lies in Equations (8) and (11), which propose to approximate the Wasserstein distance using a linear regression model on various sliced Wasserstein (SW) and lifted SW distances. While this formulation is conceptually interesting, I have several reservations about its contribution and computational benefits.

**1.1 Computational complexity**
The claimed time complexity is $O(MKL n(\log n + d))$, where $n$ is the average number of support points per distribution, $M$ is the number of sampled distribution pairs, $K$ is the number of SW variants used, and $L$ is the number of slicing directions.
However, this is not necessarily more efficient than established OT approximations. For instance, Sinkhorn OT has a complexity of $O(n^2 / \varepsilon)$ and can exploit GPU parallelism. It is therefore unclear in which practical regime the proposed regression estimator is computationally faster or more scalable. A more quantitative runtime comparison would be needed.

**1.2 Lack of transport plan**
The proposed approach only estimates the **transportation cost**, not the **transport plan**. Hence, it cannot support downstream applications that depend on the coupling (e.g., barycenters, alignment, gradient flows). This limitation makes it less general than classical OT accelerations such as Sinkhorn or low-rank OT methods. If the goal is to compute the Wasserstein distance for a single pair of distributions, this regression-based method would likely not outperform existing solvers.

**1.3 Missing comparison to linear OT**
If the intended use case is computing pairwise Wasserstein distances among many distributions, it would be natural to compare against **Linear Optimal Transport** (see [Bunne et al., 2020, “Linear Optimal Transport”](https://arxiv.org/abs/2008.09165)), which precomputes a reference map and achieves $\mathcal{O}(C(K,2) n + K n^3)$ complexity for $K$ distributions, where O(n^3) can be improved to o(n^2/\epsilon) if we use Sinkhorn to construct the embedding. Such a baseline would help clarify the advantages (if any) of the proposed regression-based approach for large-scale pairwise computation.

1.4
I suggest including **direct runtime and accuracy comparisons** against classical OT accelerations such as **Sinkhorn** and **Linear OT** in two regimes:
(1) a single pair computation, and
(2) pairwise computations among many distributions.

---

**2. Minor technical and presentation issues**

**2.1 Typographical error**
Equation (13) in Appendix A.1 appears to have a notational error: the term $\omega^{\top} S_p^{(k)\top}$ is not correctly written, as both $\omega$ and $S_p^{(k)}$ are vectors.

**2.2 Unclear expression in Figure 1**
Figure 1 may be misleading. $W_p(\mu,\nu)$ is a scalar value, not a vector; similarly, each $S_p^{(k)}(\mu,\nu)$ is also a scalar. Thus, describing $W_p$ as an “L2 projection” onto the “span” of the $S_p^{(k)}$ values is conceptually vague, since the term *span* is not well-defined in this scalar regression context. The authors may consider rephrasing this illustration or providing an intuitive geometric explanation.

**2.3 Limited theoretical analysis**
While the paper provides clean closed-form regression formulas, it lacks theoretical characterization of the estimator’s bias, variance, or generalization error. Since the model relies on a finite sample of distribution pairs, it would strengthen the paper to analyze how the regression error propagates when predicting Wasserstein distances for unseen distributions.

---

**Overall comment:**
The idea of regressing Wasserstein distances on sliced variants is creative and potentially useful for few-shot estimation, but the **computational advantage, theoretical justification, and empirical comparisons** to existing OT approximations remain insufficient. The paper would benefit significantly from a clearer positioning against established methods (Sinkhorn, linear OT) and from a more rigorous analysis of its approximation properties.

**Questions:**

### **Questions**

**Q1. Notation confusion around $P(\mu,\nu)$ in Eq. (7)**
The notation of $P(\mu,\nu)$ is confusing. From Eq. (7), it appears that $P(\mu,\nu)$ represents the *meta-distribution* used to define the expectation in Eq. (8). In this case, $\mu,\nu$ should be dummy random variables drawn from that meta-distribution. However, as currently written, $P(\mu,\nu)$ seems to be defined as a function of $\mu,\nu$ themselves, which are undefined at this point. Could the authors please clarify this notation and its relation to the sampling process in practice?

**Q2. Sampling and training details**
How are the training pairs $(\mu_i, \nu_i)$ sampled for regression fitting?
- Are they generated synthetically (e.g., Gaussian mixtures) or drawn from actual datasets (e.g., ShapeNet, MNIST)?
- What is the typical number of samples $M_0$ used for estimating regression coefficients, and how sensitive are results to $M_0$?
Providing more details would help assess reproducibility and stability.

**Q3. Choice of sliced directions ($L$) and stability**
The paper fixes $L$ as the number of slicing directions for each SW computation. How sensitive are the regression results to the choice of $L$?
In high dimensions, random projections can yield high-variance estimates of SW distances—does the regression mitigate or amplify this variance?

---

> ### Author Response · Authors · 2025-11-19
> **Response to Reviewer mmm4 (1)**
>
> First, we would like to thank the reviewer for a lot of insightful comments and questions. We would like to extend the discussion as follows:
>
> **Q1.** Computational complexity
> The claimed time complexity is $O(MKLn(\log n + d))$, where $n$ is the average number of support points per distribution, $M$ is the number of sampled distribution pairs, $K$ is the number of SW variants used, and $L$ is the number of slicing directions.
> However, this is not necessarily more efficient than established OT approximations. For instance, Sinkhorn OT has a complexity of $O(n^2 / \epsilon)$ and can exploit GPU parallelism. It is therefore unclear in which practical regime the proposed regression estimator is computationally faster or more scalable. A more quantitative runtime comparison would be needed.
>
>
> **A1.**  As discussed in the global rebuttal **GA1**, we consider the setting of computing Wasserstein distances for multiple pairs of distributions. Sinkhorn OT is a standard approximation technique for OT and Wasserstein distance for single-pair level. Therefore, Sinkhorn OT is not a direct competitor of our proposed framework. In fact, we can also use Sinkhorn (or low-rank OT) for our estimation of the regression model where we need to compute Wasserstein distances for training samples. In terms of time complexity, we split the computation into two steps: training step and inference step. In the training step, the computation of our approach is the same as the coventional Wasserstein distance since we need to compute Wasserstein distances (or their approximation e.g., via Sinkhorn) as the response for estimating the regression model. However, in the inference step, we can estimate the Wasserstein distances by applying the estimated regression function on the corresponding set of predictors (e.g., SW variants) which have lower computational complexities. In particular, the time complexity for the inference step is $\mathcal{O}((N-M)KLn(\log n + d))$ compared to $\mathcal{O}( (N-M) n^3 \log n +dn^2)$ of Wasserstein distances. Moreover, our approach is also parallelable  i.e., estimating Wasserstein distances of pairs independently given the estimated regression model. We refer the reviewer to **GA2** for a detail time and performance comparison with Sinkhorn. Overall, we see that our proposed methods leads to much better time computation than Sinkhorn in multiple-pairs setting.
>
>
>
>
>
>
> **Q2.** Lack of transport plan
> The proposed approach only estimates the **transportation cost**, not the **transport plan**. Hence, it cannot support downstream applications that depend on the coupling (e.g., barycenters, alignment, gradient flows). This limitation makes it less general than classical OT accelerations such as Sinkhorn or low-rank OT methods. If the goal is to compute the Wasserstein distance for a single pair of distributions, this regression-based method would likely not outperform existing solvers.
>
>
> **A2.** Thank you very much for your insightful comments. We agree that that proposed approach targets only the transportation cost. In principle, we can also extend the regression framework to predict the transportation plan e.g., transport plan or the corresponding Kantorovich potentials can be the responses, and sliced transport plan or their corresponding Kantorovich potentials [3] can be the predictors. However, defining a regression model on such spaces (joint distributions or functions) and the corresponding regression loss is nontrival. Therefore, we defer this to future works since we belive this direction deserve a careful investigation. In addition, there are some post-processing way to obtain a transport map from a transport cost e.g., using the Monge gap [4]. Again, this discussion is for the case of multile-pairs of distributions.
>
>
> [3] Tanguy et al, "Sliced Optimal Transport Plans"
>
> [4] Uscidda  et al, "The Monge Gap: A Regularizer to Learn All Transport Maps"

---

> > ### Author Response · Authors · 2025-11-19
> > **Response to Reviewer mmm4 (2)**
> >
> > **Q3.** Missing comparison to linear OT. If the intended use case is computing pairwise Wasserstein distances among many distributions, it would be natural to compare against **Linear Optimal Transport** (see Bunne et al., 2020, *“Linear Optimal Transport”*), which precomputes a reference map and achieves $\mathcal{O}(C(K, 2)n + K n^3)$ complexity for $K$ distributions, where $\mathcal{O}(n^3)$ can be improved to $\mathcal{O}(n^2 / \epsilon)$ if we use Sinkhorn to construct the embedding. Such a baseline would help clarify the advantages (if any) of the proposed regression-based approach for large-scale pairwise computation.
> >
> > **A3.** While we agree that the motivation of LOT and our proposed method is the same i.e., computing Wasserstein distances for multiple pairs of distributions, there are some reasons that we believe LOT is also not a direct competior of our approach. First, LOT reduces the pairwise computaton of $N$ distributions ($N^2$ OT problems between pairs) into only computing $N$ OT problems between distributions and a reference distribution. LOT provides embeddings which can be useful in other downstream tasks. However, LOT stills requires computing all $N$ OT problems, which is expensive. Moreover, for new distributions (a new pair), we need to again compute OT to the chosen reference distribution. It is worth noting that LOT requires the existence of optimal transport maps and might need a appropiate choice of the reference distribution in practice.
> >
> >
> > In contrast, our regression approach is based on estimating a model from only a subset of pairs of distributions $M<<N$. After estimation, we can perform prediction of Wasserstein distances for any pair of distributions given their sliced Wasserstein variants which are chosen for prediction. There are two benefits: we improve the computation for inference, and there is not requirement for the existence of optimal transport maps and reference distribution. We again refer the reviewer to **GA2** for a comparison with LOT where we show that our regression approach is faster and provide better estimation. For obtaining embedding of distributions, we can use our proposed regression approach to train Wasserstein Wormhole, resulting in RG-Wormhole, as discussed in the paper.
> >
> > Thank you for your pointing out LOT. We will cite it and include the discussion in the revision.
> >
> > **Q4.** I suggest including **direct runtime and accuracy comparisons** against classical OT accelerations such as **Sinkhorn** and **Linear OT** in two regimes:
> > 1. a single pair computation, and
> > 2. pairwise computations among many distributions.
> >
> > **A4.** We thank the reviewer for the constructive suggestion. We again refer the reviewer to **GA2** for a detailed comparison with Sinkhorn and Linear OT. Overall, we believe that our proposed approach is novel and practically useful for multiple-pair settings, as it addresses several limitations of Linear OT.
> >
> >
> > **Q5.** Typographical error
> > Equation (13) in Appendix A.1 appears to have a notational error: the term $\omega^\top S_p^{(k)^\top}$ is not correctly written, as both $\omega$ and $S_p^{(k)}$ are vectors.
> >
> >
> > **A5.** Thank you for pointing out the typo. We have fixed it in the revision.

---

> ### Author Response · Authors · 2025-11-19
> **Response to Reviewer mmm4 (3)**
>
> **Q6.** Unclear expression in Figure 1
> Figure 1 may be misleading. $W_p(\mu, \nu)$ is a scalar value, not a vector; similarly, each $S_p^{(k)}(\mu, \nu)$ is also a scalar. Thus, describing $W_p$ as an “L2 projection” onto the “span” of the $S_p^{(k)}$ values is conceptually vague, since the term *span* is not well-defined in this scalar regression context. The authors may consider rephrasing this illustration or providing an intuitive geometric explanation.
>
>
> **A6.** Thank you for pointing out. Figure 1 is for illustrate the sample version with the design matrix $\hat{S}$ (stacked of multiple predictors across pairs) and the response vector $\hat{W}$ (stacked of respones accross pairs). We have revised the figure in the revsion.
>
> **Q7.** Limited theoretical analysis
> While the paper provides clean closed-form regression formulas, it lacks theoretical characterization of the estimator’s bias, variance, or generalization error. Since the model relies on a finite sample of distribution pairs, it would strengthen the paper to analyze how the regression error propagates when predicting Wasserstein distances for unseen distributions.
>
> **A7.** Thank you very much for your insightful comment. There are two potential sources of randomness in our setting. First, $P(\mu, \nu)$ which represents the randomness of $\mu$ and $\nu$. Second, $\epsilon$ which represents the randomness of the regression model. If we remove the randomness of $P(\mu, \nu)$ (assuming distributions are fixed), we end up with a conventional regression setting. Since we set up of proper regression model with a linear instance, well-studied theoretical results from linear model  holds. It is well-known that for linear model, we have $\mathbb{E}[\|\omega-\hat{\omega}\|_2^2] = \sigma^2 Trace((\hat{S}^\top \hat{S})^{-1})$  which reduces approximately with the rate of $1/M$ [5] (the exact finte rate depends on the construction of $\epsilon$).   If we include the randomness of $P(\mu, \nu)$, theoretical studies become non trivial since it involves random distributions and how the randomness propagate through optimal transport distances. Therefore, we leave this investigation to future work.
>
> [5]  Greene "Econometric Analysis" Chapter 4.
>
>
> **Q8.** Notation confusion around $P(\mu, \nu)$ in Eq. (7)
> The notation of $P(\mu, \nu)$ is confusing. From Eq. (7), it appears that $P(\mu, \nu)$ represents the *meta-distribution* used to define the expectation in Eq. (8). In this case, $\mu, \nu$ should be dummy random variables drawn from that meta-distribution. However, as currently written, $P(\mu, \nu)$ seems to be defined as a function of $\mu, \nu$ themselves, which are undefined at this point. Could the authors please clarify this notation and its relation to the sampling process in practice?
>
> **A8.** Thank you very much for your comment. As answered, $P(\mu, \nu)$ presents the randomness of $\mu$ and $\nu$. Second, $\epsilon$ which represents the randomness of the regression model. We include  $P(\mu, \nu)$ for a generalization. For example, we can generate $\mu$ and $\nu$ from two Dirichlet Processes (independent or dependent) then compute sliced Wasserstein variants as predictors, combine them linearly, then add a regression noise to obtain the response which assumes to be the Wasserstein distance. Again, this hypothetical model can be misspecified. In the paper, we do not put any assumption for $P(\mu, \nu)$. However, we believe to access if a regression model is well-specified (e.g., linear model), a strong assumption on $P(\mu,\nu)$ should be posed which is also non-trivial at the moment.

---

> > ### Author Response · Authors · 2025-11-19
> > **Response to Reviewer mmm4 (4)**
> >
> > **Q9.** Sampling and training details
> > How are the training pairs $(\mu_i, \nu_i)$ sampled for regression fitting?
> >
> > - Are they generated synthetically (e.g., Gaussian mixtures) or drawn from actual datasets (e.g., ShapeNet, MNIST)?
> > - What is the typical number of samples $M_0$ used for estimating regression coefficients, and how sensitive are results to $M_0$?
> >
> > Providing more details would help assess reproducibility and stability.
> >
> > **A9.** We thank the reviewer for this question. For each experiment, the training pairs are drawn from actual datasets. Because the underlying meta-distribution $P(\mu,\nu)$ varies across datasets, the estimated model may not always transfer well between them. However, when the training and testing pairs are not substantially different, the model can still produce reasonable predictions. We refer the reviewer to **GA3**, where we trained RG on intra-class pairs and tested on inter-class pairs (and vice versa) across five scenarios on ShapeNetV2. Overall, while some performance degradation is observed, it remains moderate.
> >
> > Regarding the number of training pairs $M_0$, we typically use a small $M_0$ to estimate the regression weights in our experiments. As shown in Figures 6–13, where we vary $M_0 \in {10, 50, 100, 200}$, the model already achieves strong performance (around $R^2 \approx 0.9$) with $M_0 = 10$, and the performance improves steadily as $M_0$ increases. This suggests that the results are not overly sensitive to the exact choice of $M_0$ once it is moderately large, and that even a small training set is sufficient for the model to converge to a good estimator. Overall, we treat the choice of $M_0$ as a budget-constrained decision:  we use the largest feasible $M_0$ ratio that fits within our computational budget.
> >
> >
> > **Q10.** Choice of sliced directions ($L$) and stability
> > The paper fixes $L$ as the number of slicing directions for each SW computation. How sensitive are the regression results to the choice of $L$?
> > In high dimensions, random projections can yield high-variance estimates of SW distances - does the regression mitigate or amplify this variance?
> >
> > **A10.** We refer the reviewer to **GA4** for the sensitivity analysis regarding the number of slicing directions. In summary, as $L$ increases, the optimal weights stabilize and the performance becomes both better and more consistent. For high-dimensional datasets, we select $L$ large enough to ensure that the sliced distances are accurately estimated before performing regression. For example, on the MERFISH Cell Niches dataset (254-dimensional embeddings) we use $L = 1{,}000$ slicing directions, and on the scRNA-seq Atlas dataset (2,500 dimensions) we use $L = 10{,}000$. Overall, variance reduction for SW is an active area of research [6,7,8]. Such techniques could further improve regression quality. However, we view our proposed regression method and variance reduction for SW as largely orthogonal directions, and we leave their integration for future work.
> >
> > [6] Nguyen et al, "Sliced Wasserstein Estimation with Control Variates"
> >
> > [7] Leluc et al, "Sliced-Wasserstein Estimation with Spherical Harmonics as Control Variates"
> >
> > [8] Sisouk  et al, "A User's Guide to Sampling Strategies for Sliced Optimal Transport"

---

### Author Response · Authors · 2025-11-19
**Global Rebuttal (1)**

We sincerely thank the reviewers for their time and constructive feedback. In this global rebuttal, we summarize the common points of discussion and present new experiments. Detailed responses to individual comments can be found in the respective threads for each reviewer.

**GQ1** Relative literature position of the proposed regression approach to Sinkhorn optimal transport, linear optimal transport [2], Wasserstein Wormhole, and sliced optimal transport metrics.


**QA1** We would like to classify the computation of Wasserstein distance into two types: single-pair setting and multiple-pairs setting.


For the single-pair setting, **regularized optimal transport** is the most used approach that genuinely *approximates* the original optimal transport problem and thus yields an **approximation** to the Wasserstein distance. In contrast, **sliced optimal transport (SOT)** metrics provide **alternative** efficient distances. They are *not* approximations of the Wasserstein distance and are not generally equivalent to it [1]. Although many SOT variants exist, none are designed to **approximate** the Wasserstein distance; most yield either lower or upper bounds, as discussed in the paper.

For the multiple-pairs setting (e.g., $N$ pairs of distributions, pairwise distances of $N$ distributions), **LOT** [2] requires computing $2N$ ($N$ for pairwise distances setting) optimal transport maps to a reference distribution in order to embed distributions as functions. This relies on the existence of optimal transport maps, which typically holds only for the 2-Wasserstein distance. Another embedding approach, **Wasserstein Wormhole**, uses deep neural networks to map empirical distributions into finite-dimensional embeddings. These networks are trained iteratively using $N$ Wasserstein distance computations. Compared to LOT, Wormhole does not require the existence of transport maps and can accommodate any ground metric. LOT is also sensitive to the choice of reference distribution (which must be continuous in theory), and for any new out-of-sample distribution it must compute a new optimal transport map—typically super-cubic in the sample size $n$. In contrast, Wormhole only requires encoding the distribution with the trained networks, usually quadratic in $n$. Thus, Wasserstein Wormhole is often the most practical approach. Wasserstein distances and transport maps involved in training can also be approximated using regularized OT to reduce complexity to quadratic in $n$.

We propose a new approach for the multiple-pairs setting by formulating a principled **regression framework** (as noted by Reviewer **txMr**: *“The method is elegant, principled, and provides a practical way to ‘bootstrap’ a collection of cheap estimates to learn a high-quality one.”*). Specifically, we regress Wasserstein distances onto predictors chosen to be sliced Wasserstein and lifted sliced Wasserstein distances for computational efficiency (other distances or approximations such as Sinkhorn can also be used). The regression model is estimated using only $M << N$ distribution pairs, for which both Wasserstein distances and predictors are computed. The remaining $N-M$ Wasserstein distances are then predicted from their predictors, which can be computed in nearly linear time in $n$ (typically $n \log n$). The Wasserstein distance for any new pair can be predicted similarly. Like Wormhole, our approach is free of structural assumptions. Furthermore, we show that Wormhole can also be trained using our **predicted** Wasserstein distances to obtain embeddings. Beyond efficiency, formulating Wasserstein computation as a regression task creates a useful bridge between the regression and computational OT communities, enabling further analyses such as hypothesis testing, confidence intervals, prediction intervals, and more.

[1] Kitagawa  et al "Sliced optimal transport: is it a suitable replacement?"

[2] Moosmüller et al "Linear Optimal Transport Embedding: Provable Wasserstein classification for certain rigid transformations and perturbations"

---

> ### Author Response · Authors · 2025-11-19
> **Global Rebuttal (2)**
>
> **GQ2**  Additional experiments to compare the proposed approach to Sinkhorn optimal transport and linear optimal transport.
>
> **GA2**   As suggested by Reviewer **mmm4**, we conducted **direct runtime and accuracy comparisons** between our method (RG-s) and two widely used classical OT accelerations, **Sinkhorn** and **Linear OT**, under both single-pair computation and large-scale pairwise computations. For Sinkhorn and Linear OT, no training phase is required. For RG-s, we estimate the optimal regression weight using 5 randomly selected samples (M=10 training pairs). The training cost (2.891s) is not included in the inference-time table below, as we focus on inference time only. For Sinkhorn, we conduct a grid search over entropic regularization hyperparameter in {0.01, 0.05, 0.1} and iterations in {500, 1000}, and report the best-performing configuration for each dataset. We do not exceed 1,000 iterations because the runtime already surpasses that of exact Wasserstein.
>
> **Inference Runtime Comparison.** We use point clouds from **ShapeNetV2**, where each object consists of **2,048 points in 3D**, and measure inference time as the number of pairwise computations increases. It is worth noting that these time measurements are relative and may not be precise because of system conditions.
>
>
> | Method                     | 2 Samples (1 pair) | 10 Samples (45 pairs) | 50 Samples (1,225 pairs) | 100 Samples (4,950 pairs) | 200 Samples (19,900 pairs) |
> | -------------------------- | ------------------ | --------------------- | ------------------------ | ------------------------- | -------------------------- |
> | **Wasserstein**            | 0.604s         | 13.351s               | 354.823s                 | 1,427.276s                | 5,030.966s                 |
> | **Sinkhorn (iters=500)**   | 0.441s         | 10.444s               | 286.203s                 | 1,154.826s                | 4,628.393s                 |
> | **Sinkhorn (iters=1,000)** | 0.482s         | 21.012s               | 572.441s                 | 2,308.735s                | 9,336.621s                 |
> | **Linear OT**              | 0.634s         | 3.141s                | 15.473s                  | 31.121s                   | 92.134s                    |
> | **RG-s**                   | 0.002s        | 0.090s            | 2.043s               | 8.105s                | 21.408s                |
> | **RG-e**                  | **0.0005s**           | **0.023s**               | **1.737s**                    | **7.060s**                     | **18.429s**                     |
> | **RG-o**                  | 0.013s            | 0.136s                | 6.634s                   | 15.310s                   | 54.900s                    |
> | **RG-se**                 | 0.002s           | 0.113s               | 3.780s                    | 15.170s                    | 39.950s                     |
> | **RG-seo**                | 0.013s            | 0.147s                | 7.017s                   | 28.840s                   | 94.200s                    |
>
> **Performance Comparison.** We evaluate each method’s approximation quality relative to exact Wasserstein using $R^2$. Each value is computed on 5,000 test pairs across four datasets: MNIST point clouds, ShapeNetV2, MERFISH cell niches, and scRNA-seq.
>
> | Method    | MNIST Point Cloud | ShapeNetV2 | MERFISH Cell Niches  | scRNA-seq |
> |-------|--------|-------|--------|----|
> | **Sinkhorn**    | -1.970            | -1.549     | 0.764                | 0.964     |
> | **Linear OT**      | 0.697    | 0.918      | 0.842  | 0.984     |
> | **RG-s** | 0.925  | 0.942  | 0.964            | 0.989 |
> | **RG-e**    | 0.910     | 0.919  | 0.961    | 0.991 |
> | **RG-o**    | 0.767 | 0.753  | 0.809   | 0.923 |
> | **RG-se**  | 0.933  | 0.948  | 0.984 | **0.996** |
> | **RG-seo** | **0.934**   | **0.948**  | **0.969**            | 0.992 |
>
>
> From the results above, several observations are clear.
> (1) Sinkhorn performs the worst among all compared methods. Even with 1,000 iterations, it fails to converge on MNIST point clouds and ShapeNetV2 (negative $R^2$), while also being slower than both Linear OT and RG-s. However, this could be due to non-optimal choice of entropic regularization hyperparameter. In theory, the time complexity of Sinkhorn is $\mathcal{O}((N-M)^2 n^2/\epsilon)$ for $(N-M)^2$ pairs, hence, it is clear that Sinkhorn is slow.
>
> (2) Linear OT provides reasonable accuracy and moderate speed, but its approximation quality remains consistently below that of RG-s, and its runtime grows much faster as the number of distributions increases. As dicussed, the time complexity of LOT is $\mathcal{O}((N-M) n^2/\epsilon)$.
> (3) RG methods provide more accurate estimates than both Sinkhorn and LOT. In addition, RG-s, RG-e, RG-o, and RG-seo are all faster than LOT. RG-seo achieves a speed comparable to LOR while offering better approximation quality (whereas optimization-based SW variants require substantially more computation time).

---

> > ### Author Response · Authors · 2025-11-19
> > **Global Rebuttal (3)**
> >
> > **GQ3.** Generalization across intra-class and inter-class pairs.
> >
> > **GA3.** Following reviewer **AafT**’s suggestion,  we also conducted an experiment to test how well RG generalizes across intra-class and inter-class pairs. We trained RG only on intra-class pairs and tested on inter-class pairs, and vice versa, using 5 scenarios for each setting on ShapeNetV2. Although this causes intentionally suboptimal (the regression weight is learned from a restricted subset of pairs), RG still achieves reasonably high $R^2$ (e.g., 0.902, 0.846, 0.932, 0.805 in the reported cases), compared to about 0.95 when training on the full pair distribution. This shows that **RG generalizes well for out-samples data**, with performance degradation that is present but moderate. The full results are reported in **Figures 29–30 in the Appendix**.
> >
> > **GQ4.** How sensitive is the learned regressor to the choice and number of slicing directions ($L$)?
> >
> > **GA4.** As suggested by reviewers **mmm4** and **AafT**, we conducted a sensitivity study to assess how the choice of $L$ affects the performance of RG. We used the RG-s instance on 3D point clouds, fixed the dataset, and varied $L \in \{1,5,10,50,100\}$. For each value of $L$, we refit the regression ten times. The table below shows that as we increase the number of projections, the optimal weights become more concentrated (their variance decreases) and the $R^2$ against the exact Wasserstein distance is substantially improved and also becomes more stable across runs.
> >
> > | $L$ | $w_0$  | $w_1$  | $R^2$         |
> > | --- | ----- | ---- | ---|
> > | 1   | 0.605 ± 0.215 | 0.463 ± 0.063 | 0.693 ± 0.070 |
> > | 5   | 1.202 ± 0.319 | 0.284 ± 0.096 | 0.815 ± 0.058 |
> > | 10  | 1.427 ± 0.203 | 0.212 ± 0.050 | 0.895 ± 0.064 |
> > | 50  | 1.284 ± 0.209 | 0.230 ± 0.061 | 0.924 ± 0.018 |
> > | 100 | 1.400 ± 0.122 | 0.188 ± 0.028 | 0.947 ± 0.019 |
> >
> >  As expected, increasing $L$ makes these sliced features more stable and systematically improves the regression fit.
> >
> > **GQ5.** Regression Coefficients visualization.
> >
> > **GA5.** We visualized the estimated coefficients of the RG methods for all applications in **Figures 21–28 in the Appendix**. The coefficient patterns vary substantially across datasets, indicating that there is **no fixed set of coefficients** that universally applies to all cases. This further underscores the need for the proposed regression framework, which captures the dataset-specific uncertainty in the association between Wasserstein and sliced Wasserstein distances (i.e., the meta-distribution).
> >
> > **GQ6.** Training time comparison between RG vs. Wormhole.
> >
> > **GA.** As suggested by reviewer **A3f7**, We report experiments where RG and Wormhole are trained on the same number of training samples and compare their training time under this matched setting. In both cases, we must compute Wasserstein distances for a set of sample pairs. For RG, these distances are computed once to serve as regression targets (together with the corresponding sliced Wasserstein–type distances), and the regression weight is then obtained by a closed-form solution, which adds negligible overhead. In contrast, Wormhole repeatedly computes pairwise Wasserstein distances at every training iteration (twice per iteration for the encoder and decoder losses) while simultaneously updating the parameters of a Transformer-based backbone, resulting in substantially higher runtime.
> >
> > To further highlight this efficiency gap, we conducted an additional experiment on a point-cloud dataset where each object consists of 2,048 3D points. We compare the training time of RG variants with the baseline Wormhole. Wormhole is trained for only 10 epochs under identical settings, even though it typically requires around **1,000 epochs** with a batch size of 20 to converge. The results are summarized below:
> >
> > | Method    | 10 Samples (45 pairs) | 50 Samples (1,225 pairs)| 100 Samples (4,950 pairs) | 200 Samples (19,900 pairs) |
> > |-----|----|-----|---|-----|
> > | **RG-s** | 14.202s  | 383.991s   | 1,516.343s  | 5,127.080s  |
> > | **RG-e** | 14.072s| 383.390s   | 1,512.727s  | 5,123.932s  |
> > | **RG-o**  | 16.355s | 427.079s   | 1,685.878s  | 5,823.599s  |
> > | **RG-se** | 14.362s    | 385.740s   | 1,519.995s  | 5,155.342s  |
> > | **RG-seo** | 18.432s    | 429.575s   | 1,688.742s  | 5,863.439s  |
> > | **Wormhole (10 epochs)**   | 384.683s   | 3,296.464s | 7,348.263s  | 14,022.843s |
> >
> > These results demonstrate that RG variants provide **over an order of magnitude speed-up** compared to Wormhole, and the gap widens as the number of training samples increases.
> >
> > In practice, RG variants are linear models, so only a **small number of training samples** is sufficient to estimate the regression weight. As shown in **Figures 6–13 in Appendix B.4**, RG already achieves high performance when the training set size is 10, and the performance then improves steadily as the training size increases, indicating that **training with a few samples is already close to convergence** for our model.

---

### Meta-Review · Area_Chair_2Vqk · 2026-01-06

**Summary:**

The framework for approximating Wasserstein distances through regression on sliced Wasserstein (SW) is useful.

The missing points (e.g., training time, complexity) were well addresses during rebuttal.

**Reviewer Concerns:**

The efficiency of the method was presented. The need of samples remains unclear.

**Reviewer Scores:**

The weakness raised by Reviewer A3f7 were addressed but no responses from this reviewer.

---

### Decision · Program_Chairs · 2026-01-26

Accept (Poster)